# POSE PRIORS FROM LANGUAGE MODELS

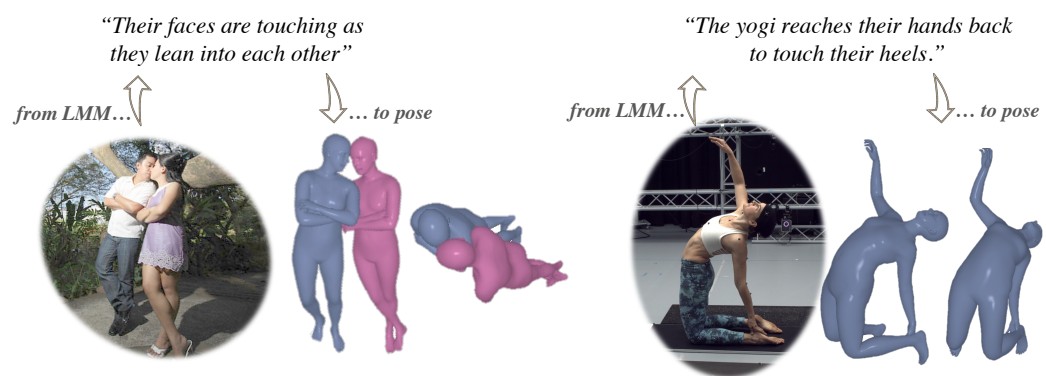

*"Their faces are touching as they lean into each other"*

*"The yogi reaches their hands back to touch their heels."*

*from LMM…*      *… to pose*      *from LMM…*      *… to pose*

Figure 1: **Optimizing contacts in 3D human pose.** Our approach leverages the semantic priors of a Large Multimodal Model (LMM) by converting natural language descriptions of individuals in an image into mathematical constraints. We can then optimize the 3D pose estimates using these constraints. These examples show image descriptions generated by an LMM and corresponding refined pose estimates.

## ABSTRACT

We present a pose optimization method that enforces accurate physical contact constraints when estimating the 3D pose of humans. Our central insight is that since language is often used to describe physical interaction, large pretrained text-based models can act as priors on pose estimation. We can thus leverage this insight to improve pose estimation by converting natural language descriptors, generated by a large multimodal model (LMM), into tractable losses to constrain the 3D pose optimization. Despite its simplicity, our method produces surprisingly compelling pose reconstructions of people in close contact, correctly capturing the semantics of the social and physical interactions. We demonstrate that our method rivals more complex state-of-the-art approaches that require expensive human annotation of contact points and training specialized models. Moreover, unlike previous approaches, our method provides a unified framework for resolving self-contact and person-to-person contact.[1]

## 1 INTRODUCTION

In 3D pose estimation, the dominant forms of labeled training data are 3D pose (obtained via motion capture) and 2D keypoints (Goel et al., 2023). Scaling the amount of this data is expensive, as it requires either special technology or fine-grained human effort. On the other hand, in 2D computer vision tasks such as recognition and segmentation, prior work shows that natural language supervision provides a path to strong performance (Radford et al., 2021; Xu et al., 2022a). Can we use language-based models to improve 3D pose estimation?

Our focus in this work is enabling computers to correctly perceive physical contact when estimating pose (See Figure 1). Specifically, we aim to build a system that takes as input a single view of people during close physical interaction or one person in a pose that involves self-contact and produces accurate 3D mesh reconstructions of each person as output. This setting is challenging for state-of-the-art pose regression models, as some body parts are frequently occluded by other ones, and also

---

[1]Our code will be publicly available at the time of publication.

challenging for pose optimization methods relying on 2D keypoints, which do not convey contact points. Previously proposed approaches address these issues by curating task-specific datasets via motion capture or human-annotated points of contact between body parts (Muller et al., 2021; Fieraru et al., 2021; Müller et al., 2023).

As physical contact is a universal human social signal, humans developed extensive terminology for its particularities. Detailed descriptions of touch in different contexts are widely discussed in texts that range from love-song lyrics such as Paul Anka's "Put your head on my shoulder" to Shakespeare's "See how she leans her cheek upon her hand." (Romeo and Juliet). It touches on subjects from love to meditative poses.

Our main insight is that since written language discusses our physical interactions (hugs, kisses, fist fights, yoga poses, etc.) at great length, we should be able to extract a semantic prior on humans' poses from a pretrained large multimodal model (LMM) (Achiam et al., 2023; Liu et al., 2023; Dai et al., 2023). Just like a prior trained on motion capture data, this language-based prior can tell us which contacts are most likely in poses and interactions. Through this approach, we avoid the time-consuming and expensive collection of training data involving motion capture or annotated self and cross-person contacts that previous refinement methods require.

This insight leads us to a simple framework, which we call ProsePose. We prompt a pre-trained LMM, with the image and request as output a formatted list of contact constraints between body parts. We then convert this list of constraints into a loss function that can be optimized jointly with other common losses, such as 2D keypoint loss, to refine the initial estimates of a pose regression model. The prompt provides an intuitive way for the system designer to adapt the generated constraints to their setting (e.g. if they want to focus on yoga or dance).

We show in experiments on three 2-person interaction datasets and one dataset of complex yoga poses that ProsePose produces more accurate reconstructions than previous approaches that do not use a large amount of task-specific data for training. These results indicate that LMMs, without any additional finetuning, offer a useful prior for pose reconstruction.

In summary, (1) we show that LMMs have implicit semantic knowledge of poses that is useful for pose estimation, and (2) we formulate a novel framework that converts free-form natural language responses from a pre-trained LMM into tractable loss functions that can be used for pose optimization.

## 2 RELATED WORK

### 2.1 3D HUMAN POSE RECONSTRUCTION

Reconstructing 3D human poses from single images is an active area of research. Prior works have explored using optimization-based approaches (Pavlakos et al., 2019a; Guan et al., 2009; Lassner et al., 2017; Pavlakos et al., 2019b; Rempe et al., 2021) or pure regression (Kanazawa et al., 2018; Arnab et al., 2019; Guler & Kokkinos, 2019; Joo et al., 2021; Kolotouros et al., 2019) to estimate the 3D body pose given a single image. HMR2 (Goel et al., 2023) is a recent state-of-the-art regression model in this line of work. Building on these monocular reconstruction approaches, some methods have looked into reconstructing multiple individuals jointly from a single image. These methods (Zanfir et al., 2018; Jiang et al., 2020; Sun et al., 2021) use deep networks to reason about multiple people in a scene to directly output multi-person 3D pose predictions. BEV (Sun et al., 2022) accounts for the relative proximity of people explicitly using relative depth annotations to reason about proxemics when predicting and placing each individuals in the scene (e.g. depth of people with respect to one another). However, approaches in both categories generally do not accurately capture physical contact between parts of a single person or between people (Müller et al., 2023; Muller et al., 2021).

### 2.2 CONTACT INFERENCE IN 3D POSE RECONSTRUCTION

3D pose reconstruction is especially challenging when there is self-contact or inter-person contact. This has motivated a line of work on pose reconstruction approaches tailored for this setting. Muller et al. (2021) focuses on predicting self contact regions for 3D pose estimation by leveraging a dataset with collected contact annotations to model complex poses such as arm on hip or crossed arms. Fieraru et al. (2020) introduces the first dataset with hand-annotated ground truth contact labels between two people. REMIPS (Fieraru et al., 2021) and BUDDI (Müller et al., 2023) train models on the person-to-person contact maps in this data in order to improve 3D pose estimation of multiple people from a single image. CloseInt (Huang et al., 2024) trains a physics-guided diffusion model on

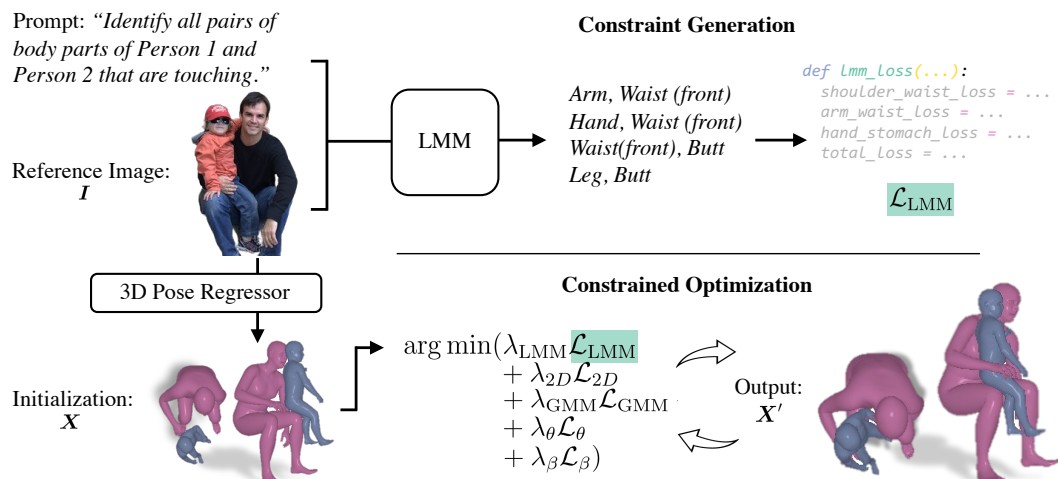

Figure 2: **LMM-guided Pose Estimation** Our method takes as input an image of one or two people in contact. We first obtain initial pose estimates for each person from a pose regressor. Then we use an LMM to generate contact constraints, each of which is a pair of body parts that should be touching. This list of contacts is converted into a loss function $\mathcal{L}_{\text{LMM}}$. We optimize the pose estimates using $\mathcal{L}_{\text{LMM}}$ and other losses to produce a refined estimate of each person's pose that respects the predicted contacts.

two-person motion capture data for this task. However, contact annotations, which are crucial for these approaches, are difficult and expensive to acquire. Our method does not require any training on such annotations. Instead, we leverage an LMM's implicit knowledge about pose to constrain pose optimization to capture both self- and person-to-person contact.

## 2.3 LANGUAGE PRIORS ON HUMAN POSE

There exists a plethora of text to 3D human pose and motion datasets (Punnakkal et al., 2021; Guo et al., 2022; Plappert et al., 2016), which have enabled work focused on generating 3D motion sequences of a single person performing a general action (Tevet et al., 2023; Jiang et al., 2023; Zhang et al., 2023). This line of work has been extended to generating the motion of two people conditioned on text (Shafir et al., 2023; Liang et al., 2023).

PoseScript (Delmas, Ginger and Weinzaepfel, Philippe and Lucas, Thomas and Moreno-Noguer, Francesc and Rogez, Grégory, 2022) is a method for generating a single person's pose from fine-grained descriptions. They leverage a library of predefined pose descriptors, from which they form detailed textual annotations for their motion capture dataset. By training a model on this data, they can generate various plausible poses. PoseFix (Delmas, Ginger and Weinzaepfel, Philippe and Moreno-Noguer, Francesc and Rogez, Grégory, 2023) considers the problem of modifying a pose given a fine-grained description of the desired change, and introduces a labeled dataset for this task. The PoseFix method then trains a model on this data to predict the modified pose given the initial pose and description. PoseGPT (Feng et al., 2023), like our work, focuses on the problem of monocular 3D reconstruction of people. PoseGPT is a pose regressor that uses language as part of its training data. However, PoseGPT does not produce better pose estimates than previous state-of-the-art regressors (i.e. regressors that do not use language) and applies only to the one-person setting.

Our work differs from previous work on language and pose in several ways. First, whereas all prior work trains a model on data with pairs of language and pose, which is expensive to collect, our method leverages the existing knowledge in an LMM to reason about pose. Second, prior work in this area focuses on either the one-person or the two-person setting. In contrast, our work presents a single framework to reason about physical contacts within or between poses. Finally, in scenes with physical contact, we show that our method improves the pose estimates of state-of-the-art regressors.

## 3 GUIDING POSE OPTIMIZATION WITH AN LMM

Given an image, our goal is to estimate the 3D body pose of individuals in the image while capturing the self and cross-person contact points. While we cannot trivially use natural language responses (hug, kiss) to directly optimize 3D body poses, we leverage the key insight that LMMs understand *how* to articulate a given pose (arms around waist, lips touching). We propose a method to structure these articulations into constraints and convert them into loss functions.

More concretely, our framework, illustrated by Figure 2, takes as input the image $\boldsymbol{I}$ and the bounding boxes $\boldsymbol{B}$ of the subjects of interest. In the first stage, the image is passed to a pose regressor to obtain a rough estimate of the 3D pose $\boldsymbol{X}^p$ for each individual $p$ in the image. In the second stage, we prompt a LMM with the image and a set of instructions in order to generate a list of self- or inter-person contact constraints, which we then convert into a loss function (Sec. 3.4). Finally, in the third stage, we jointly optimize the generated loss function with several other pre-defined loss terms (Sec. 3.4). We refer to our framework as **ProsePose** .

### 3.1 PRELIMINARIES

While our approach scales in principle to an arbitrary number of individuals, we focus our description on the two-person case to keep the exposition simple. We also demonstrate results on the one-person case, which is simply an extension of the two-person case. In particular, we apply our method to the one-person case by setting $X^0 = X^1$. Please see § 6 for details on the differences between the two-person and one-person cases.

**Large Multimodal Models** An LMM is a model that takes as input an image and a text prompt and produces text output that answers the prompt based on the image. Our framework is agnostic to the architecture of the LMM. LMMs are typically trained to respond to wide variety of instructions (Liu et al., 2023; Dai et al., 2023), but at the same time, LMMs are prone to hallucination (Leng et al., 2023; Li et al., 2023). Handling cases of hallucination is a key challenge when using LMMs, and we mitigate this issue by aggregating information across several samples from the LMM.

**Pose representation.** We use a human body model (Pavlakos et al., 2019a) to represent each person $p \in \{0, 1\}$. The body model is composed of a pose parameter that defines the joint rotations $\boldsymbol{\theta} \in \mathbb{R}^{d_\theta \times 3}$, where $d_\theta$ is the number of joints, and a shape parameter $\boldsymbol{\beta} \in \mathbb{R}^{d_\beta}$, where $d_\beta$ is the dimensions of the shape parameter. We can apply a global rotation $\boldsymbol{\Phi} \in \mathbb{R}^3$ and translation $\boldsymbol{t} \in \mathbb{R}^3$ to place each person in the world coordinate space. The full set of parameters for each person is denoted by $\boldsymbol{X}^p = [\boldsymbol{\theta}^p, \boldsymbol{\beta}^p, \boldsymbol{\Phi}^p, \boldsymbol{t}^p]$. For simplicity, we refer to the parameter set $(\boldsymbol{X}^0, \boldsymbol{X}^1)$ as $\boldsymbol{X}$.

These parameters can be plugged into a differentiable function that maps to a mesh consisting of $d_v$ vertices $\boldsymbol{V} \in \mathbb{R}^{d_v \times 3}$. From the mesh, we can obtain a subset of the vertices representing the 3D locations of the body's joints $\boldsymbol{J} \in \mathbb{R}^{d_j \times 3}$. From these joints, we can calculate the 2D keypoints $\boldsymbol{K}_{proj}$ by projecting the 3D joints to 2D using the camera intrinsics $\Pi$ predicted from (Pavlakos et al., 2019a).

$$\boldsymbol{K}_{proj} = \Pi\left(\boldsymbol{J}\right) \in \mathbb{R}^{d_j \times 2}. \tag{1}$$

**Vertex regions.** In order to define contact constraints between body parts, we define a set of *regions* of vertices. Prior work on contact has partitioned the body in to fine-grained regions (Fieraru et al., 2020). However, since our constraints are specified by a LMM trained on natural language, the referenced body parts are often coarser in granularity. We therefore update the set of regions to reflect this language bias by combining these fine-grain regions into larger, more commonly referenced body parts such as arm, shoulder (front&back), back, and waist (front&back). Please see § 6.2 for a visualization of the coarse regions. Formally, we write $\boldsymbol{R} \in \mathbb{R}^{d_r \times 3}$ to denote a region with $d_r$ vertices, which is part of the full mesh ($\boldsymbol{R} \subset \boldsymbol{V}$).

**Constraint definition.** A contact constraint specifies which body parts from two meshes should be touching. Using the set of coarse regions, we define contact constraints as pairs of coarse regions $\boldsymbol{c} = (\boldsymbol{R}_a, \boldsymbol{R}_b)$ between a region $\boldsymbol{R}_a$ of one mesh and $\boldsymbol{R}_b$ of the other mesh, as shown in Figure 3. For instance, ("hand", "arm") indicates a hand should touch an arm.

### 3.2 POSE INITIALIZATION

We obtain a rough initial estimate of the 3D pose from a regression-based method. The regressor takes as input the image $\boldsymbol{I}$ and outputs estimates for the body model parameters $\boldsymbol{\theta}, \boldsymbol{\beta}, \boldsymbol{r}$, and $\boldsymbol{t}$ for each subject.

### 3.3 CONSTRAINT GENERATION WITH A LMM

Our method strives to enforce contact constraints for the estimated 3D poses. Our key insight is to leverage a LMM to identify regions of contact between different body parts on the human body surface. As shown in Figure 2 (top), we prompt the LMM with an image and ask it to output a list of

all plausible regions that are in contact. However, we cannot simply use natural language descriptions to directly optimize a 3D mesh. As such, we propose a framework to convert these constraints into a loss function.

**LMM-based constraint generation.** Given the image $I$, we first use the bounding boxes $B$ to crop the part containing the subjects. We then use an image segmentation model to mask any extraneous individuals. While cropping and masking the image may remove information, we find the LMMs are relatively robust to missing context, and more importantly, this allows us to indicate which individuals to focus on. Given the segmented image, we ask the LMM to generate a set $C = \{c_1, ... c_m\}$ of all pairs of body parts that are touching, where $m$ is the total number of constraints the LMM generates for the image.

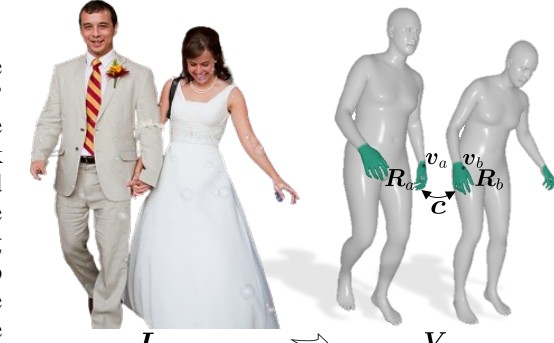

Figure 3: **Notation.** Given an image $I$, we can lift each individual into corresponding 3D meshes $V$. We define contact constraints $c$ as pairs of regions $(R_a, R_b)$ in contact. The loss is defined in terms of the distance between the vertices $(v_a, v_b)$ on the mesh.

In the prompt, we specify the full set of coarse regions to pick from. We find that LMMs fail to reliably reference the left and right limbs correctly or consistently, so we designed this set of coarse regions such that they do not disambiguate the chirality of the hands, arms, legs, feet, and shoulders. Instead, the two hands are grouped together, the two arms are grouped together, etc. Nevertheless, if the LMM uses "left" or "right" to reference a region, despite the instruction to not do so, we directly use the part of the region with the specified chirality rather than considering both possibilities.

Motivated by the chain-of-thought technique, which has been shown to improve language model performance on reasoning tasks (Wei et al., 2022), we ask the LMM to write its reasoning or describe the pose before listing the constraints. For the full prompt used in each setting, please refer to § 6.

We sample $N$ responses from the LMM, yielding $N$ sets of constraints $\{C_1, C_2, ..., C_N\}$. The next step is to convert each constraint set $C_j$, where $j \in \{1, 2, ... N\}$, into a loss term.

**Loss function generation.** We first filter out contact pairs that occur fewer than $f$ times across all constraint sets, where $f$ is a hyperparameter. Then for each contact pair $c = (R_a, R_b)$ in $C_j$, we define $dist(c)$ as the minimum distance between the two regions:

$$dist(c) = \min \|v_a - v_b\|_2 \quad \forall v_a \in R_a, \forall v_b \in R_b \tag{2}$$

where $\{v_a, v_b\} \in \mathbb{R}^3$. In practice, the number of vertices in each region can be very large. To make this computation tractable, we first take a random sample of vertices from $R_a$ and from $R_b$ before computing distances between pairs of vertices in these samples. Furthermore, since the ordering of the people in the LMM constraints is unknown (i.e. does $R_a$ come from the mesh defined by parameter $X^0$ or $X^1$), we compute the overall loss for both possibilities and take the minimum. We use $c^\top = (R_b, R_a)$ to denote the flipped ordering. We then sum over all constraints in the list $C_j$:

$$dist_{\text{sum}}(C_j) = \min \left( \sum_{c \in C_j} dist(c), \sum_{c \in C_j} dist(c^\top) \right) \tag{3}$$

Each constraint set sampled from the LMM is likely to contain noise or hallucination. To mitigate the effect of this, we average over all $N$ losses corresponding to each constraint set to obtain the overall LMM loss. This technique is similar to self-consistency (Wang et al., 2022), which is commononly used for code generation tasks. Concretely, the overall LMM loss is defined as

$$\mathcal{L}_{\text{LMM}} = \frac{1}{N} \sum_{j=1}^{N} dist_{sum}(C_j) \tag{4}$$

If a constraint set $C_j$ is empty (i.e. the LMM does not suggest any contact pairs), then we set $dist_{\text{sum}}(C_j) = 0$. If there are several such constraint sets, we infer that the LMM has low confidence about the contact points (if any) in the image. To handle these cases, we set a threshold $t$ and if the number of empty constraint sets is at least as large as $t$, we gracefully backoff to the appropriate baseline optimization procedure (described in Sections 4.1 and 4.2 for each setting).

### 3.4 CONSTRAINED POSE OPTIMIZATION

Drawing from previous optimization-based approaches (Müller et al., 2023; Bogo et al., 2016; Pavlakos et al., 2019b), we employ several additional losses in the optimization. We then minimize the joint loss to obtain a refined subset of the body model parameters $X' = [\theta', \beta', t']$:

$$[\theta', \beta', t'] = \arg\min(\lambda_{\text{LMM}}\mathcal{L}_{\text{LMM}} + \lambda_{\text{GMM}}\mathcal{L}_{\text{GMM}} + \lambda_\beta\mathcal{L}_\beta + \lambda_\theta\mathcal{L}_\theta + \lambda_{2D}\mathcal{L}_{2D} + \lambda_P\mathcal{L}_P)$$

Following Müller et al. (2023), we divide the optimization into two stages. In the first stage, we optimize all three parameters. In the second stage, we optimize only $\theta$ and $t$, keeping the shape $\beta$ fixed. Here, we detail all of the remaining losses used in the optimization.

**Pose and shape priors.** We compute a loss $\mathcal{L}_{\text{GMM}}$ based on the Gaussian Mixture pose prior of Bogo et al. (2016) and a shape loss $\mathcal{L}_\beta = \|\beta\|_2^2$, which penalizes extreme deviations from the body model's mean shape.

**Initial pose loss.** To ensure we do not stray too far from the initialization, we penalize large deviations from the initial pose $\mathcal{L}_\theta = \|\theta' - \theta\|_2^2$.

**2D keypoint loss.** Similar to BUDDI (Müller et al., 2023), for each person in the image, we obtain pseudo ground truth 2D keypoints and their confidences from OpenPose (Cao et al., 2019) and ViTPose (Xu et al., 2022b). Given this pseudo ground truth, we merge all the keypoints into $K \in \mathbb{R}^{d_j \times 2}$, and their corresponding confidences into $\gamma \in \mathbb{R}^{d_j}$. From the predicted $X'$, we can compute the 2D projection of each 3D joint location using Equation 3.1. Then, the 2D keypoint loss is defined as:

$$\mathcal{L}_{2D} = \sum_{j=1}^{d_j} \gamma(K_{proj} - K)^2 \tag{5}$$

**Interpenetration loss.** To prevent parts of one mesh from being in the interior of the other, we add an interpenetration loss. Generically, given two sets of vertices $V_0$ and $V_1$, we use winding numbers to compute the subset of $V_0$ that intersects $V_1$, which we denote as $V_{0,1}$. Similarly, $V_{1,0}$ is the subset of $V_1$ that intersects $V_0$. The interpenetration loss is then defined as

$$\mathcal{L}_P = \sum_{x \in V_{0,1}} \min_{v_1 \in V_1} \|x - v_1\|_2^2 + \sum_{y \in V_{1,0}} \min_{v_0 \in V_0} \|y - v_0\|_2^2 \tag{6}$$

Due to computational cost, this loss is computed on low-resolution versions of the two meshes (roughly 1000 vertices per mesh).

## 4 EXPERIMENTS

We conduct experiments on several datasets in the two-person and one-person settings. In this section, we first provide important implementation details and a description of the metrics that we use to evaluate our method and previous approaches. We then present quantitative and qualitative results showing that ProsePose refines pose estimates to capture semantically relevant contact in each setting.

**Implementation details.** Following prior work on two-person pose estimation (Müller et al., 2023), we use BEV (Sun et al., 2022) to initialize the poses since it was trained to predict both the body pose parameters and the placement of each person in the scene. However, on the single person yoga poses, we find that the pose parameter estimates of HMR2 (Goel et al., 2023) are much higher quality, so we initialize the body pose using HMR2.

We use the SMPL-X (Pavlakos et al., 2019a) body model and GPT4-V (Achiam et al., 2023) as the LMM with temperature = 0.7 when sampling from it.[2] We also include results when using LLaVA as the LMM in § 7.4. We use Segment Anything (Kirillov et al., 2023) as the segmentation model, used to remove extraneous people in the image (we only apply this step for FlickrCI3D, since other datasets are from motion capture). Unless otherwise specified, we set $N = 20$ samples. For all

---

[2]We access GPT4-V, specifically the `gpt-4-vision-preview` model, via the OpenAI API: platform.openai.com. We use the "high" detail setting for image input.

Table 1: **Two-person Results.** Joint PA-MPJPE (lower is better) and Avg. PCC (higher is better). For FlickrCI3D, PA-MPJPE is computed using the pseudo-ground-truth fits. **Bold** indicates best method without contact supervision in each column.

| | Hi4D | FlickrCI3D | | CHI3D | |
| | PA-MPJPE$_\downarrow$ | PA-MPJPE$_\downarrow$ | PCC$_\uparrow$ | PA-MPJPE$_\downarrow$ | PCC$_\uparrow$ |
|---|---|---|---|---|---|
| *Without contact supervision* | | | | | |
| BEV (Sun et al., 2022) | 144 | 106 | 64.8 | **96** | 71.4 |
| Heuristic | 116 | 67 | 77.8 | 105 | 74.1 |
| ProsePose | **93** | **58** | **79.9** | 100 | **75.8** |
| *With contact supervision* | | | | | |
| BUDDI (Müller et al., 2023) | 89 | 65.9 | 81.9 | 68 | 78.6 |

of our 2-person experiments, $f = 1$, while $f = 10$ in the 1-person setting. We set $t = 2$ for the experiment on the CHI3D dataset and $t = N$ for all other experiments. We set $\lambda_{\text{LMM}} = 1000$ in the 2-person experiments, and $\lambda_{\text{LMM}} = 10000$ in the 1-person setting. In the two-person case, all other loss coefficients are taken directly from Müller et al. (2023). In the one-person case, we find that removing the GMM pose prior and doubling the weight on the initial pose loss improves optimization dramatically, likely because the complex yoga poses are out of distribution for the GMM prior. These hyperparameters and our prompts were chosen based on experiments on the validation sets. Furthermore, following Müller et al. (2023), we run both optimization stages for at most 1000 steps. We use the Adam optimizer (Kingma & Ba, 2014) with learning rate 0.01. For other implementation details such as prompts, the list of coarse regions in each setting, and additional differences between the 1- and 2-person cases, please refer to § 6.

**Metrics.** As is standard in the pose estimation literature, we report Procrustes-aligned Mean Per Joint Position Error (PA-MPJPE) in millimeters. This metric finds the best alignment between the estimated and ground-truth pose before computing the joint error. In the two-person setting, we focus on the *joint* PA-MPJPE, as this evaluation incorporates the relative translation and orientation of the two people. See § 7.2 for the per-person PA-MPJPE.

We also include the percentage of correct contact points (PCC) metric introduced by (Müller et al., 2023). This metric captures the fraction of ground-truth contact pairs that are accurately predicted. For a given radius $r$, a pair is classified as "in contact" if the two regions are both within the specified radius. We use the set of fine-grained regions defined in Fieraru et al. (2020) to compute PCC. The metric is averaged over $r \in 0, 5, 10, 15, ..., 95$ mm. Please note that since these regions are defined on the SMPL-X mesh topology, we convert the regression baselines– BEV and HMR2– from the SMPL mesh topology to SMPL-X to compute this metric. Please see § 7.1 for more details on the regions and on the mesh conversion.

### 4.1 TWO-PERSON POSE REFINEMENT

**Datasets** We evaluate on three datasets, and our dataset processing largely follows (Müller et al., 2023). **Hi4D** (Yin et al., 2023) is a motion capture dataset of pairs of people interacting. Each sequence has a subset of frames marked as contact frames, and we take every fifth contact frame. We use the images from a single camera, resulting in roughly 247 images. **Flickr Close Interactions 3D (FlickrCI3D)** (Fieraru et al., 2020) is a collection of Flickr images of multiple people in close interaction. The dataset includes manual annotations of the contact maps between pairs of people. (Müller et al., 2023) used these contact maps to create pseudo-ground truth 3D meshes and curated a version of the test set to exclude noisy annotations, which has roughly 1403 images. **CHI3D** (Fieraru et al., 2020) is a motion capture dataset of pairs of people interacting. We present results on the validation set. There are 126 different sequences, each of which has a single designated "contact frame." Each frame is captured from 4 cameras, so there are roughly 504 images in this set.

To develop our method, we experimented on the validation sets of FlickrCI3D and Hi4D, and a sample of the training set from CHI3D. For our experiments, we can compute the PCC on FlickrCI3D and CHI3D, which have annotated ground-truth contact maps. Since all baselines also use BEV for initialization, we exclude images where BEV fails to detect one of the subjects in the interaction pair.

**Baselines** We compare our estimated poses to the following:

.

Table 2: **Two-person PCC.** Percent of correct contact points (PCC) for five different radii $r$ in mm. **Bold** indicates the best score wothout contact supervision in each column. At the ground-truth contact points, our method brings the meshes closer together than the baselines.

| | PCC$_\uparrow$ @ $r$ on FlickrCI3D | | | | | PCC$_\uparrow$ @ $r$ on CHI3D | | | | |
|---|---|---|---|---|---|---|---|---|---|---|
| | 5 | 10 | 15 | 20 | 25 | 5 | 10 | 15 | 20 | 25 |
| *Without contact supervision* | | | | | | | | | | |
| BEV (Sun et al., 2022) | 3.6 | 6.3 | 10.8 | 17.1 | 28.6 | 5.8 | 17.4 | 32.5 | 47.3 | 61.9 |
| Heuristic | 14.6 | 33.9 | 49.3 | 60.8 | 70.3 | 11.1 | 28.0 | 45.3 | 55.3 | 64.4 |
| ProsePose | **15.6** | **39.9** | **57.1** | **67.9** | **75.8** | **13.5** | **35.2** | **52.5** | **61.3** | **68.4** |
| *With contact supervision* | | | | | | | | | | |
| BUDDI (Müller et al., 2023) | 18.5 | 44.2 | 61.8 | 73.1 | 80.8 | 15.7 | 39.4 | 57.1 | 68.8 | 78.0 |

Figure 4: **Two-person results** We show qualitative results from ProsePose , BUDDI (Müller et al., 2023), and the contact heuristic. Under each example, we show the top 3 constraints predicted by GPT4-V and the number of times each constraint was predicted across all 20 samples. Our method correctly reconstructs people in a variety of interactions, and the predicted constraints generally align with the interaction type in each example.

- **BEV (Sun et al., 2022)** Multi-person 3D pose estimation method. Uses relative depth to reason about spatial placement of individuals in the scene. ProsePose , Heuristic, and BUDDI use BEV to initialize pose estimates.

- **Heuristic** A contact heuristic which includes the auxiliary losses in Section 3.4 as well as a term that minimizes the minimum distance between the two meshes. Introduced by (Müller et al., 2023). We use their hyperparameters for this heuristic. Please note, this baseline is used as the default when the number of empty constraint sets is at least the threshold $t$.

- **BUDDI (Müller et al., 2023)** This method uses a learned diffusion prior to constrain the optimization. We stress that BUDDI requires a large amount of annotated training data on pairs of interacting bodies, which is not used in our method.

**Quantitative Results** Table 1 provides quantitative results on the three datasets.

Across datasets, ProsePose consistently improves over the strongest baseline, **Heuristic**.

On the Hi4D dataset, ProsePose reduces 85% of the gap in PA-MPJPE between **Heuristic** and the fully supervised **BUDDI**. On the FlickrCI3D and CHI3D datasets, ProsePose narrows the gap in the average PCC between **Heuristic** and **BUDDI** by more than one-

third. (While ProsePose achieves a better PA-MPJPE than **BUDDI** on FlickrCI3D, for this dataset, we rely primarily on PCC since PA-MPJPE is computed on *pseudo*-ground-truth fits.)

On CHI3D, ProsePose outperforms **Heuristic** but underperforms **BEV** in terms of PA-MPJPE. We find that on the subset of images where we do not default to the heuristic (i.e. on images where GPT4-V predicts enough non-empty constraint sets), the PA-MPJPE for ProsePose and BEV is 86 and 87, respectively. In other words, in the cases where our method is actually used, the joint error is slightly less than that of BEV. As a result, we can attribute the worse overall error to the poorer performance of the heuristic. Overall, our method improves over the other methods that do not use 3D supervision in terms of both joint error and PCC. Table 2 shows the PCC for each method at various radii. The results show that ProsePose brings the meshes closer together at the correct contact points. On both the FlickrCI3D and CHI3D datasets, ProsePose outperforms the other baselines that do not use contact supervision.

Next, we ablate important aspects of ProsePose . In Figure 5, we show that averaging the loss over several samples from the LMM improves performance, mitigating the effect of LMM hallucination. Table 3 presents an ablation of all the losses involved in our optimization on the Hi4D validation set. $\mathcal{L}_{\text{LMM}}$ and $\mathcal{L}_{2D}$ have the greatest impact, indicating that our LMM-based loss is crucial for the large improvement in joint error.

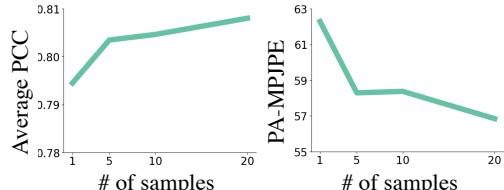

Figure 5: **More samples improve pose estimation.** On the FlickrCI3D validation set, taking more samples from the LMM and averaging the resulting loss functions improves joint PA-MPJPE (left) and average PCC (right).

|  | PA-MPJPE↓ |
|---|---|
| All Losses | 81 |
| w/o. $\mathcal{L}_{\text{LMM}}$ | 138 |
| w/o. $\mathcal{L}_{\text{GMM}}$ | 85 |
| w/o. $\mathcal{L}_{\beta}$ | 91 |
| w/o. $\mathcal{L}_{\theta}$ | 84 |
| w/o. $\mathcal{L}_{2D}$ | 130 |
| w/o. $\mathcal{L}_{P}$ | 78 |

Table 3: **Ablations on Hi4D.** Joint PA-MPJPE (lower is better). We evaluate the impact of each loss in our optimization on the Hi4D by removing one loss at a time. For all experiments, we use the same settings. The set of cases where we default to the baseline (Heuristic) is also kept the same.

**Qualitative Results** Figure 4 shows examples of reconstructions from ProsePose , **Heuristic**, and **BUDDI**. Below each of our predictions, we list the most common constraints predicted by GPT4-V for the image. The predicted constraints correctly capture the semantics of each interaction. For instance, it is inherent that in tango, one person's arm should touch the other's back. In a rugby tackle, a player's arms are usually wrapped around the other player. Using these constraints, ProsePose correctly reconstructs a variety of interactions, such as tackling, dancing, and holding hands. In contrast, the heuristic struggles to accurately position individuals and/or predict limb placements, often resulting in awkward distances.

### 4.2 ONE-PERSON POSE REFINEMENT

**Datasets** Next, we evaluate ProsePose on a single-person setting. For this setting, we evaluate on MOYO (Tripathi et al., 2023), a motion capture dataset with videos of a single person performing various yoga poses. The dataset provides views from multiple different cameras. We pick a single camera that shows the side view for evaluation. For each video, we take single frame from the middle as it generally shows the main pose. There is no official test set, and the official validation set consists of only 16 poses. Therefore, we created our own split by picking 79 arbitrary examples from the training set to form our validation set. We then combine the remaining examples in the training set with the official validation set to form our test set. In total, our test set is composed of 76 examples. Since this dataset does not have annotated region contact pairs, we compute the pesudo-ground-truth contact maps using the Euclidean and geodesic distance following Muller et al. (2021).

**Baselines** We compare against the following baselines:

- **HMR2** (Goel et al., 2023) State-of-the-art pose regression method. We use this baseline to initialize our pose estimates for optimization.
- **HMR2+opt** Optimization procedure that is identical to our method without $\mathcal{L}_{\text{LMM}}$. This method is the default when the number of empty constraint sets is at least the threshold $t$.

Both the quantitative and qualitative results echo the trends discussed in the 2-person setting. Table 4 provides the quantitative results. The PCC metrics show that our LMM loss improves the predicted

.

Table 4: **One-person Results.** PA-MPJPE (lower is better) and Avg. PCC (higher is better). Our method captures ground-truth contacts better than the baseline methods, as shown by the PCC.

| | PA-MPJPE$_\downarrow$ | PCC$_\uparrow$ | PCC$_\uparrow$ @ $r$ | | | | |
| --- | --- | --- | --- | --- | --- | --- | --- |
| | | | 5 | 10 | 15 | 20 | 25 |
| HMR2 (Goel et al., 2023) | 84 | 83.0 | 34.2 | 55.2 | 69.5 | 78.4 | 83.9 |
| HMR2+opt | **81** | 85.2 | 47.7 | 65.5 | 74.6 | 80.9 | 86.2 |
| ProsePose | 82 | **87.8** | **54.2** | **73.8** | **81.4** | **86.5** | **91.3** |

Input    ProsePose    HMR2    HMR2-opt    Input    ProsePose    HMR2    HMR2-opt

*Hand, Foot* × 21                          *Hand, Hand* × 14

*Hand, Foot* × 21                          *Hand, Foot* × 18

Figure 6: **Single-person results** We show qualitative results from ProsePose , HMR2 (Goel et al., 2023), and HMR2-optim on complex yoga poses. Each example also shows the constraints that are predicted by the LMM at least $f = 10$ times (and are thus used to compute $\mathcal{L}_{\text{LMM}}$) with their counts. ProsePose correctly identifies self-contact points and optimizes the poses to respect these contacts.

self-contact in complex yoga poses relative to the two baselines. Figure 6 provides a qualitative comparison of poses predicted by ProsePose versus the two baselines. Below each of our predictions, we list the corresponding constraints predicted by GPT4-V. In each case, the predicted constraint captures the correct self-contact, which is reflected in the final pose estimates. With the addition of the semantically guided loss, ProsePose effectively refines the pose to ensure proper contact between hand-foot or hand-hand, an important detail consistently overlooked by the baselines.

### 4.3 LIMITATIONS

While ProsePose consistently improves contact across settings and datasets, it has some limitations. First, though we mitigate it through averaging, LMM hallucination of incorrect constraints may lead to an unexpected output. Second, when taking the minimum loss across the possible chiralities of limbs, the pose initialization may lead to a suboptimal choice. We show in § 7.3 examples of failure cases like these. We also note that the LMM may be biased toward poses common in certain cultures due to its training data. In addition, we find that GPT4-V performs worse with some of the camera angles in the MOYO dataset (e.g. frontal or aerial), perhaps because in photos yoga poses are most often captured from a side view.

### 5 CONCLUSION

We present ProsePose , a zero-shot framework for refining 3D pose estimates to capture touch accurately using the implicit semantic knowledge of poses in LMMs. Our key novelty is that we generate structured pose descriptions from LMMs and convert them into loss functions used to optimize the pose. Since ProsePose does not require training, we eliminate the need for the expensive contact annotations used in prior work to train priors for contact estimation. Our framework applies in principle to an arbitrary number of people, and our experiments show in both one-person and two-person settings, ProsePose improves over previous zero-shot baselines. More broadly, this work provides evidence that LMMs are promising tools for 3D pose estimation, which may have implications beyond touch.

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

## APPENDIX FOR POSE PRIORS FROM LANGUAGE MODELS

In this appendix, we provide additional details about our method (Section 6), details about metrics (Section 7.1), additional quantitative results (Section 7.2), examples of failure cases (Section 7.3), experiments with a different LMM (Section 7.4), and more qualitative comparisons (Section 7.5). We also provide a video overview of the method and qualitative results: https://drive.google.com/file/d/1blaLnALiOd4C-au8GW61CtThsolWeLf3/view?usp=sharing.

## 6 ADDITIONAL METHOD DETAILS

### 6.1 LMM PROMPTS

The box below contains our prompt for the two-person experiments.

> You are a helpful assistant. You follow all directions correctly and precisely.
> For each image, identify all pairs of body parts of Person 1 and Person 2 that are touching.
> Write all of these in a Markdown table where the first column is "Person 1 Body Part" and the second column is "Person 2 Body Part".
> You can pick which is Person 1 and which is Person 2.
> The list of possible body parts is: head, neck, chest, stomach, waist (back), waist (front), back, shoulder (back), shoulder (front), arm, hand, leg, foot, butt.
> Do not include left/right.
> List ALL pairs you are confident about.
> If you are not confident about any pairs, output an empty table.
> Carefully write your reasoning first, and then write the Markdown table.

The box below contains our prompt for the one-person experiment.

> You are a helpful assistant. You answer all questions carefully and correctly.
> Identify which body parts of the yogi are touching each other in this image (if any).
> Write each pair in a Markdown table with two columns.
> Each body part MUST be from this list:
> head, back, shoulder, arm, hand, leg, foot, stomach, butt, ground
> Do not write "left" or "right".
> Describe and name the yoga pose, and then write the Markdown table.
> Note that the pose may differ from the standard version, so pay close attention.
> Only list a part if you're certain about it.

In each setting, the prompt is given as the "system prompt" to the GPT-4 API, and the only other message given as input contains the input image with the "high" detail setting.

### 6.2 COARSE REGIONS

Figure 7 illustrates the coarse regions referenced in the prompt in our two-person experiments. Figure 8 illustrates the coarse regions referenced in the prompt in our one-person experiments. In the one-person case, the prompt does not mention the "chest," "neck," or "waist" regions, since they tend to be less important for contacts in yoga poses, and the front/back shoulders are merged into one region, since the distinction tends to be less important for contacts in yoga poses.

### 6.3 CONVERTING CONSTRAINTS TO LOSSES IN 1 VS. 2 PERSON CASES

Our implementation of the conversion from constraints output by the LMM to loss functions differs slightly between the two-person and one-person cases.

810
811
812
813
814
815
816
817
818
819
820
821
822
823
824
825
826
827
828
829
830
831
832
833

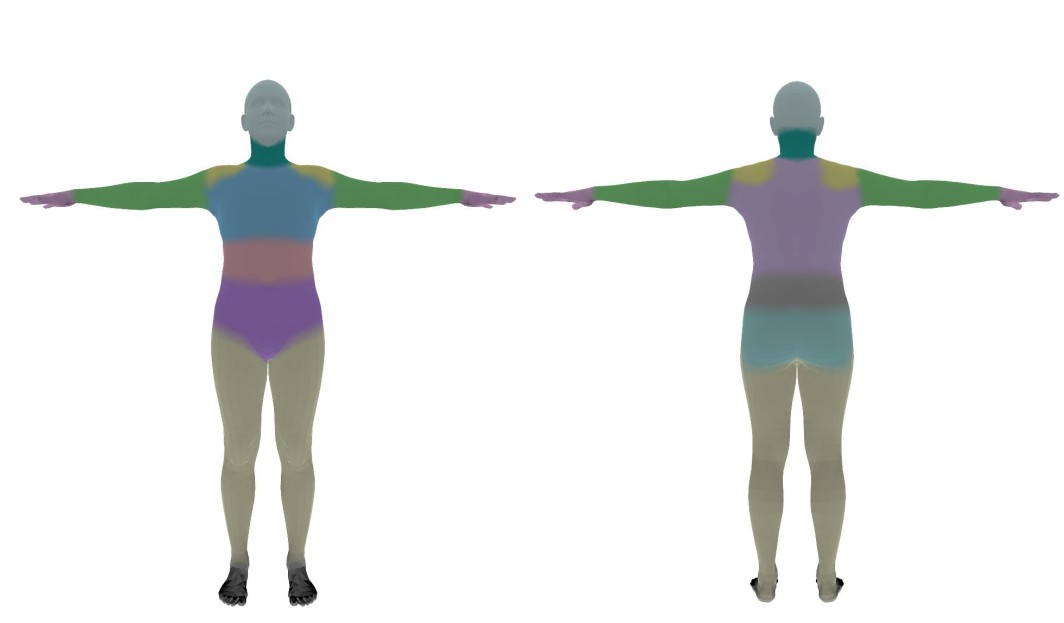

Figure 7: Color-coded coarse regions in the two-person prompt: head, neck, chest, stomach, waist (back), waist (front), back, shoulder (back), shoulder (front), arm, hand, leg, foot, butt. Note that some of these regions overlap. For instance, the "back" includes the "waist (back)" and "shoulder (back)" regions as a subset.

834
835
836
837
838
839
840
841
842
843
844
845
846
847
848
849
850
851
852
853
854
855
856
857
858
859
860
861
862
863

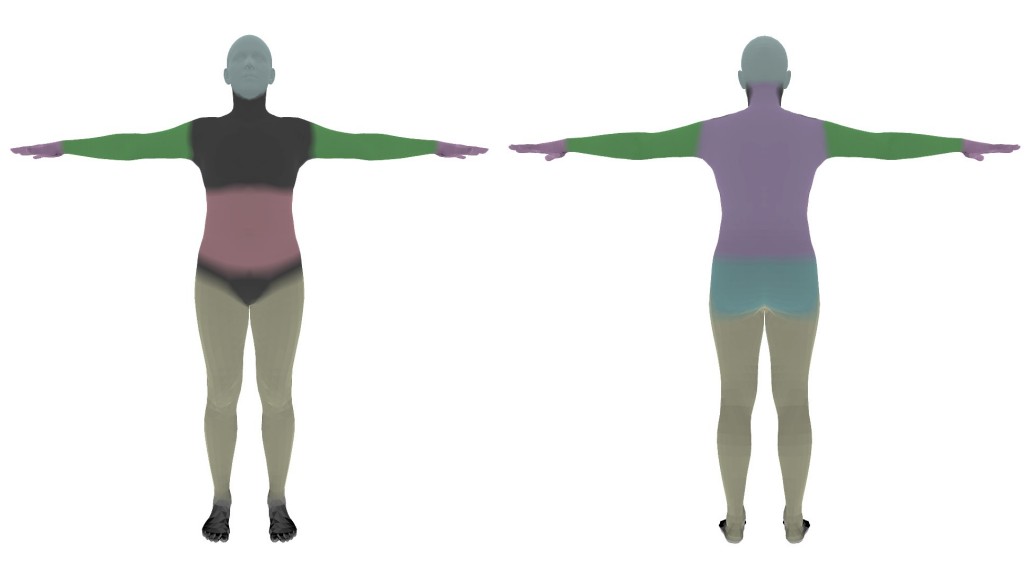

Figure 8: Color-coded coarse regions in the one-person prompt: head, stomach, back, shoulder, arm, hand, leg, foot, butt. Note that the "chest," "neck," and "waist (front)" regions are not covered by the regions in the prompt, since they tend to have less importance for contacts in yoga poses.

### 6.3.1 TWO-PERSON

Since we ask the VLM not to differentiate between "left" and "right" limbs, when there should be a constraint on both limbs (e.g. both hands), taking the minimum distance independently for each constraint pair may lead to a constraint on only one limb. Consequently, if the same body part (e.g. "hand") is mentioned in at least two separate rows of the table output by the LMM (without any "left" or "right" prefix), we enforce that both the left and right limbs of this type must participate in the loss.

We also handle some variations in how the LMM references body parts. First, we check for the following terms in addition to the coarse regions named in the prompt: left hand, right hand, left arm, right arm, left foot, right foot, left leg, right leg, left shoulder, right shoulder, left shoulder (front), right shoulder (front), left shoulder (back), right shoulder (back), waist. "waist" corresponds to the union of "waist (front)" and "waist (back)." Each of these terms is mapped to the corresponding set of fine-grained regions, similar to the coarse regions shown in Figure 7. As stated in Section 3.3 of the main paper, if a "left" or "right" part is explicitly named by the LMM's output, this part of the coarse region is directly used without considering the other part.

Second, we find there are some cases where the LMM expresses uncertainty between regions using a delimiter like "/" (e.g. "hand / arm"). So we split each entry in the Markdown Table's output by the delimiter "/" and we compute the loss for each possible region that is listed; we then sum all of these losses.

### 6.3.2 ONE-PERSON

In the one-person experiment, we do not make use of the constraints involving the "ground" that the LMM outputs. Similar to the two-person case, the code for converting the LMM's output to a loss function checks for the following terms in addition to the body regions listed in the prompt: left hand, right hand, left arm, right arm, left foot, right foot, left leg, right leg, left shoulder, right shoulder, left shoulder (front), right shoulder (front), left shoulder (back), right shoulder (back), waist . Each of these terms is mapped to the corresponding set of fine-grained regions, similar to the coarse regions shown in Figure 7.

### 6.4 BOUNDING BOXES AND CROPPING

As stated in Section 3 of the main paper, we take bounding boxes of the subjects of interest as input and use them to crop the image in order to isolate the person/people of interest when prompting the LMM. For FlickrCI3D, we use the ground-truth bounding boxes of the two subjects of interest. For the other datasets, we use keypoints detected by ViTPose/OpenPose to create the bounding boxes. For the single-person MOYO dataset, we manually check that the bounding boxes from the keypoints and the selected HMR2 outputs correspond to the correct person in the image. We note that the baseline HMR2+opt also benefits from this manual checking, since HMR2+opt also depends on the HMR2 outputs and accurate keypoints.

## 7 EXPERIMENTS

### 7.1 PCC CALCULATION

Figure 9 illustrates the 75 fine-grained regions used for PCC calculation, which are the same as those used in Fieraru et al. (2020). We opted to compute PCC on the fine-grained regions rather than on the coarse ones since prior work uses the fine-grained regions Müller et al. (2023) and since we want to measure contact correctness at a finer granularity (e.g. upper vs. lower thigh vs. knee). Since the regressors BEV and HMR2 use the SMPL mesh while the fine-grained regions are defined on the SMPL-X mesh, we use a matrix $M \in \mathbb{R}^{\text{num\_vertices\_smplx} \times \text{num\_vertices\_smpl}}$ to convert the SMPL meshes to SMPL-X in order to compute PCC.

### 7.2 PER-PERSON PA-MPJPE

Table 5 shows the per-person PA-MPJPE for each of the datasets used in our two-person experiments.

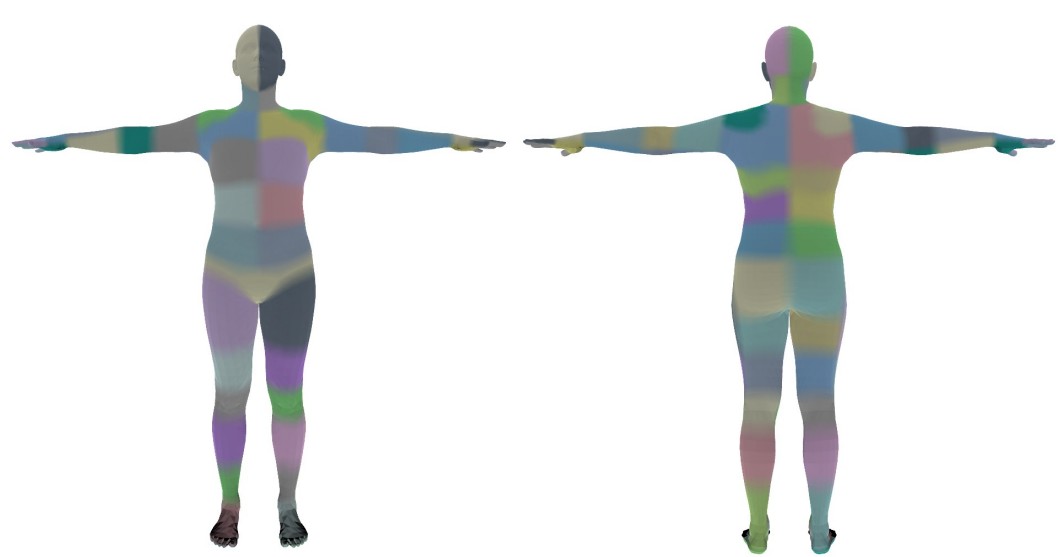

Figure 9: Color-coded 75 fine-grained regions used for PCC calculation

Table 5: **Two-person Results.** Per-person PA-MPJPE (lower is better). For FlickrCI3D, PA-MPJPE is computed using the pseudo-ground-truth fits.

|  | Hi4D PA-MPJPE↓ | FlickrCI3D PA-MPJPE↓ | CHI3D PA-MPJPE↓ |
|---|---|---|---|
| *Without contact supervision* | | | |
| BEV Sun et al. (2022) | 76 | 71 | 51 |
| Heuristic | 65 | 31 | 48 |
| ProsePose | 65 | 31 | 49 |
| *With contact supervision* | | | |
| BUDDI Müller et al. (2023) | 70 | 43 | 47 |

## 7.3 FAILURE CASES

Figure 10 shows examples of two types of LingoPose failures: (1) incorrect chirality (example a) and (2) hallucination (examples b and c). In example (a), the top constraints are correct but without the chirality specified. The optimization then brings both hands of one person to roughly the same point on the other person's waist, rather than positioning one hand on each hip. Similarly, both hands of the other person are positioned on the same shoulder of the first person. Examples (b) and (c) both show cases of hallucination. In example (b), the hand is predicted to touch the back rather than the hand. In example (c), the hand is predicted to touch the foot rather than the leg. Interestingly, in the yoga example, GPT4-V correctly predicts the name of the yoga pose in all 20 samples ("Parivrtta Janu Sirsasana"). However, it outputs a constraint between a hand and a foot, which is true in the standard form of this pose but not in the displayed form of the pose. Consequently, the optimization brings the left hand closer to the right foot than to the right knee.

## 7.4 DIFFERENT MULTIMODAL MODEL

In this section, we evaluate ProsePose when using a different LMM. We use LLaVA-NeXT 34B (i.e. LLaVA v1.6) Liu et al. (2023) as the LMM. We find that the model does not perform well in directly generating the table of constraints from the image. This is presumably a result of a weaker language model in LLaVA compared to GPT4 Therefore, we instead generate a caption from the LMM, and we feed the caption alone to GPT4 in order to convert it into a table of constraints. We evaluated a few different prompts on the validation sets and chose the prompts with the best performance therein. For the two-person experiments, we use the following prompt for LLaVA:

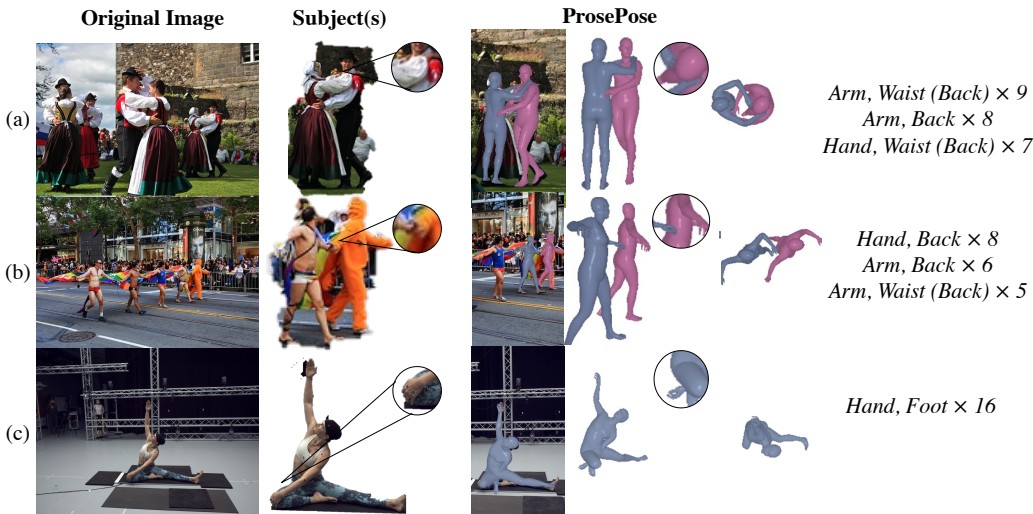

**Original Image**   **Subject(s)**   **ProsePose**

(a) *Arm, Waist (Back) × 9*
*Arm, Back × 8*
*Hand, Waist (Back) × 7*

(b) *Hand, Back × 8*
*Arm, Back × 6*
*Arm, Waist (Back) × 5*

(c) *Hand, Foot × 16*

Figure 10: **Failure cases** We show examples in which ProsePose fails to output a semantically correct pose. The constraints shown are the top 3 constraints (or the total number of constraints, whichever is smaller) that meet the threshold $f$ along with their counts ($f = 1$ for two-person experiments and $f = 10$ for the one-person experiment).

> Describe the pose of the two people.

We then use the following prompt with GPT4 to rewrite the caption so that it does not mention left and right to refer to limbs, since we find that the LMM is not reliably correct in doing so:

> Rewrite the caption below so that it doesn't mention "left" or "right" to describe any hand, arm, foot, or leg. The revised caption should otherwise be identical. Write only the revised caption and no other text.

We then use the following prompt with GPT4 to create the formatted table.

> You are a helpful assistant. You will follow ALL rules and directions entirely and precisely. Given a description of Person 1 and Person 2 who are physically in contact with each other, create a Markdown table with the columns "Person 1 Body Part" and "Person 2 Body Part", listing the body parts of the two people that are guaranteed to be in contact with each other, from the following list. ALL body parts that you list must be from this list. You can choose which person is Person 1 and which is Person 2. Body parts: "chest", "stomach", "waist (front)", "waist (back)", "shoulder (front)", "shoulder (back)", "back", "hand", "arm", "foot", "leg", "head", "neck", "butt" Note that "back" includes the entire area of the back.
> Include all contact points that are directly implied by the description, not just those that are explicitly mentioned. If there are no contact points between these body parts that the description implicitly or explicitly implies, your table should contain only the column names and no other rows.
> First, write your reasoning. Then write the Markdown table.

For the one-person case, we use the following prompt for LLaVA:

> Describe the person's pose.

We use the same prompt as above to rewrite the caption. We then use the following prompt to create the formatted table:

Table 6: **LLaVA Results.** Err denotes Joint PA-MPJPE for the two-person datasets (Hi4D, FlickrCI3D, CHI3D) and PA-MPJPE for MOYO. Lower is better for Err, and higher is better for Avg. PCC. **Bold** indicates best method without contact supervision in each column.

| | Hi4D | FlickrCI3D | | CHI3D | | MOYO | |
| | Err$_\downarrow$ | Err$_\downarrow$ | PCC$_\uparrow$ | Err$_\downarrow$ | PCC$_\uparrow$ | Err$_\downarrow$ | PCC$_\uparrow$ |
| --- | --- | --- | --- | --- | --- | --- | --- |
| Heuristic | 116 | 67 | 77.8 | 105 | 74.1 | – | – |
| HMR2+opt | – | – | – | – | – | 81 | 85.2 |
| GPT4-V | 93 | 58 | 79.9 | 100 | 75.8 | 82 | 87.8 |
| LLaVA+GPT4 | 95 | 60 | 79.7 | 101 | 75.2 | 82 | 85.2 |

> You are a helpful assistant. You will follow ALL rules and directions entirely and precisely. Given a description of a yoga pose, create a Markdown table with the columns "Body Part 1" and "Body Part 2", listing the body parts of the person that are guaranteed to be in contact with each other, from the following list. ALL body parts that you list must be from this list. Body parts: "head", "back", "shoulder", "arm", "hand", "leg", "foot", "stomach", "butt", "ground" Note that "back" includes the entire area of the back.
> Include all contact points that are directly implied by the description, not just those that are explicitly mentioned. If there are no contact points between these body parts that the description implicitly or explicitly implies, your table should contain only the column names and no other rows.
> First, write your reasoning. Then write the Markdown table.

We use the `gpt-4-0125-preview` version of GPT4 via the OpenAI API (we obtained better results using this model than `gpt-4-1106-preview`). The latency of this approach is much higher than the single-stage approach used with GPT4-V, since we must feed each caption individually to the OpenAI API. Therefore, we set $N = 5$ for these experiments. Since we change $N$, we also need to select appropriate thresholds $f$ and $t$. As in the experiments with GPT4-V, we set $t = N$ for all datasets except CHI3D. For CHI3D, we find on the validation set that $t = 2$ works better than $t = 1$, so we set $t = 2$. As in the experiments with GPT4-V, we set $f = 1$ for the 2-person datasets, and we set $f = 3$ for MOYO, to approximate the ratio $f/N$ used in the GPT4-V experiments. Finally, when converting the constraint pairs to loss functions, we found that on a small number of examples, the pipeline produced a large number of constraints, leading to very slow loss functions. Therefore, we discarded loss functions that are longer than 10000 characters.

Table 6 shows the results. On the 2-person datasets, the LLaVA+GPT4 approach performs better than the contact heuristic but not as well as GPT4-V. This is in line with holistic multimodal evaluations that indicate that GPT4-V performs better than LLaVA Lu et al. (2024). On the 1-person yoga dataset, the performance of LLaVA+GPT4 is comparable with that of the baseline (HMR2+opt). The reason that LLaVA performs worse than GPT4-V in this setting may be that LLaVA does not have enough training data on yoga to provide useful constraints.

## 7.5 ADDITIONAL QUALITATIVE RESULTS

Figures 11, 12, 13, and 14 show additional, randomly selected examples from the multi-person FlickrCI3D test set. Figures 15, 16, 17, and 18 show the same examples comparing ProsePose with the pseudo-ground truth fits. Figures 19, 20, and 21 show additional, randomly selected examples from the Hi4D test set. Figures 22 and 23 show additional, randomly selected examples from the CHI3D validation set (which we use as the test set following Müller et al. (2023)). Figures 24 and 25 show additional, randomly selected examples from the 1-person yoga MOYO test set.

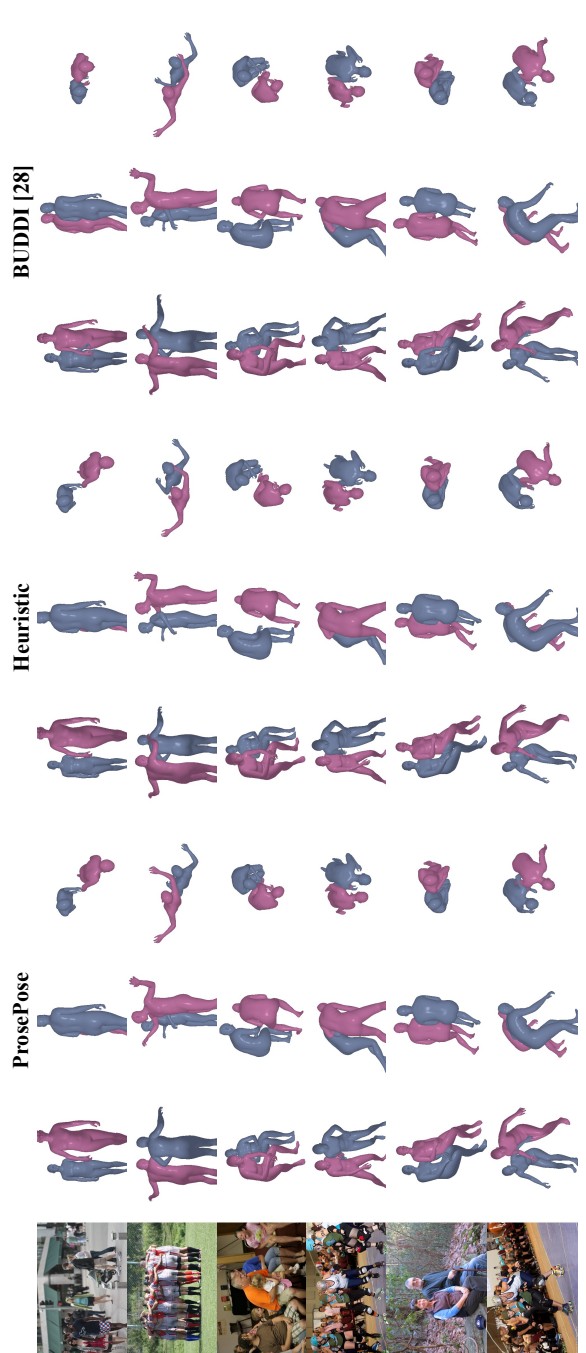

Figure 11: Non-curated examples from the FlickrCI3D test set. They are randomly selected from the examples for which there is at least one non-empty constraint set.

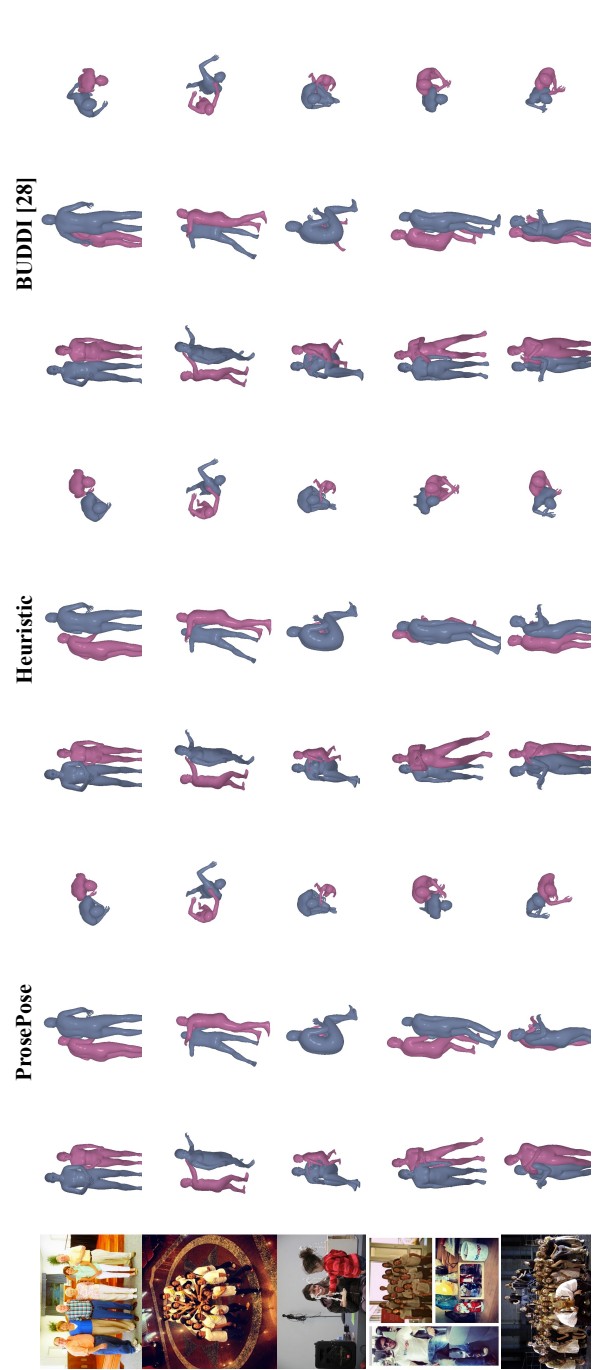

Figure 12: Non-curated examples from the FlickrCI3D test set. They are randomly selected from the examples for which there is at least one non-empty constraint set.

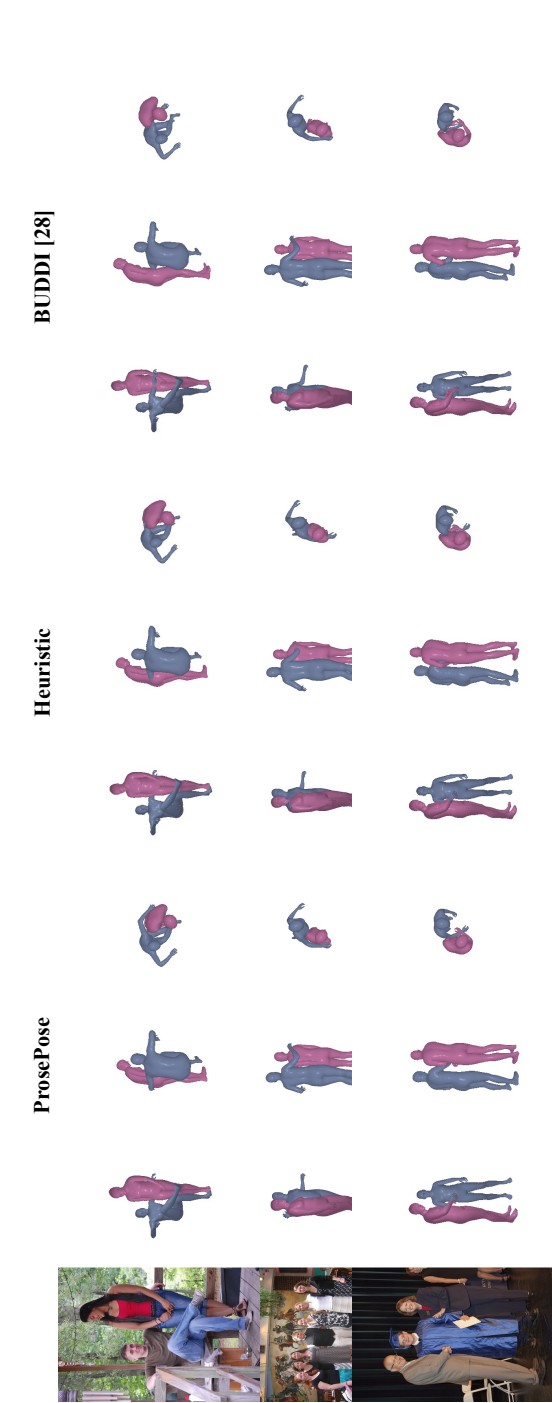

Figure 13: Non-curated examples from the FlickrCI3D test set. They are randomly selected from the examples for which there is at least one non-empty constraint set.

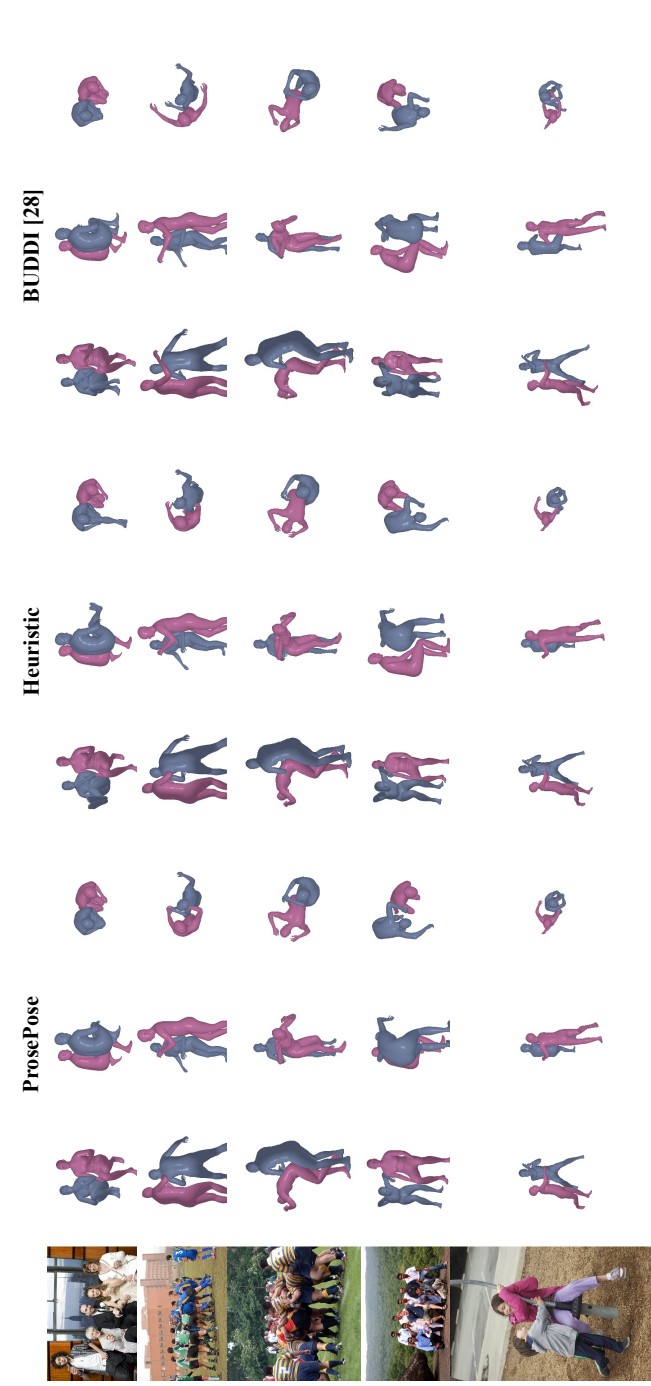

Figure 14: Non-curated examples from the FlickrCI3D test set. They are randomly selected from the examples for which there is at least one non-empty constraint set.

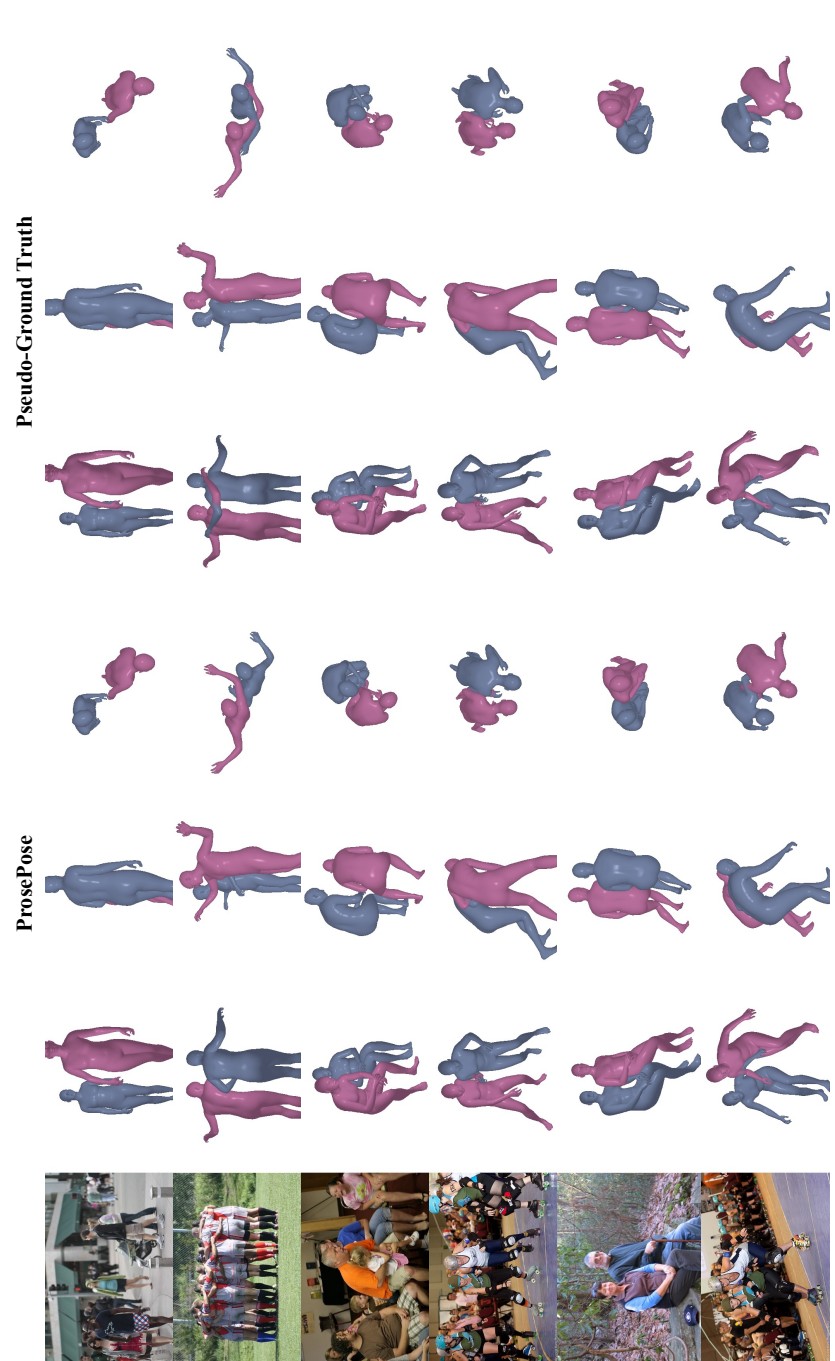

Figure 15: Non-curated examples from the FlickrCI3D test set, comparing ProsePose with the pseudo-ground truth fits. They are randomly selected from the examples for which there is at least one non-empty constraint set.

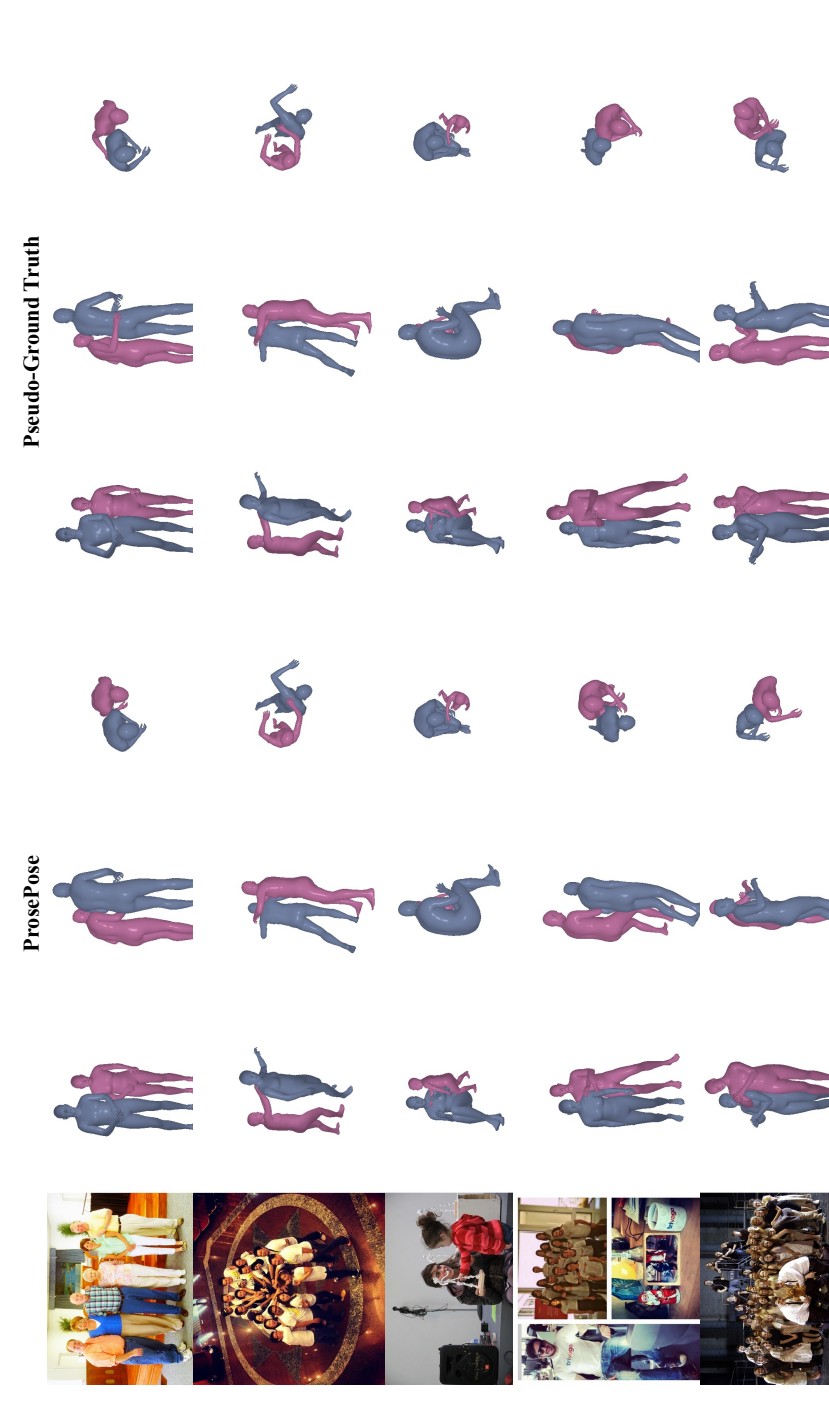

Figure 16: Non-curated examples from the FlickrCI3D test set, comparing ProsePose with the pseudo-ground truth fits. They are randomly selected from the examples for which there is at least one non-empty constraint set.

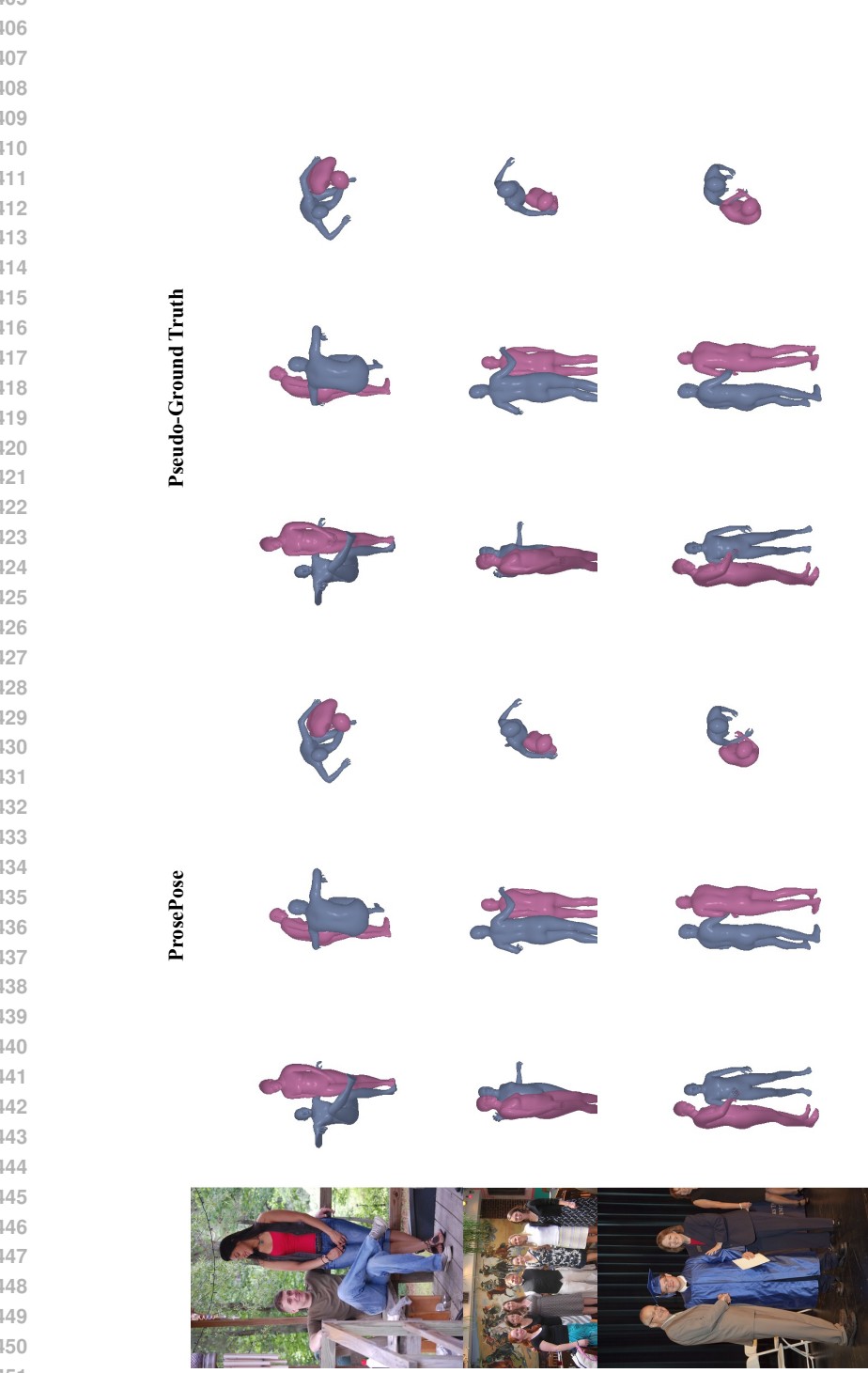

Figure 17: Non-curated examples from the FlickrCI3D test set, comparing ProsePose with the pseudo-ground truth fits. They are randomly selected from the examples for which there is at least one non-empty constraint set.

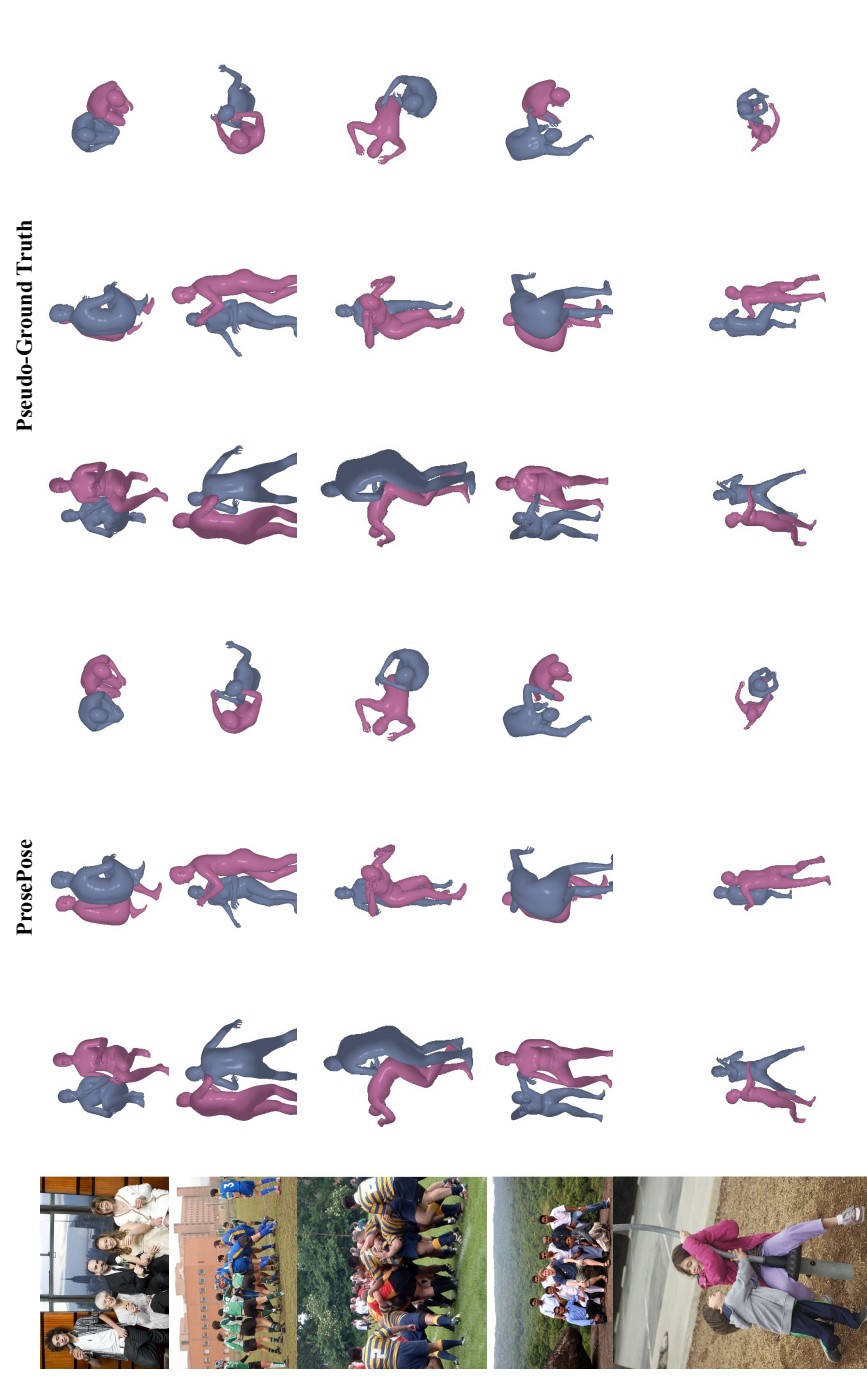

Figure 18: Non-curated examples from the FlickrCI3D test set, comparing ProsePose with the pseudo-ground truth fits. They are randomly selected from the examples for which there is at least one non-empty constraint set.

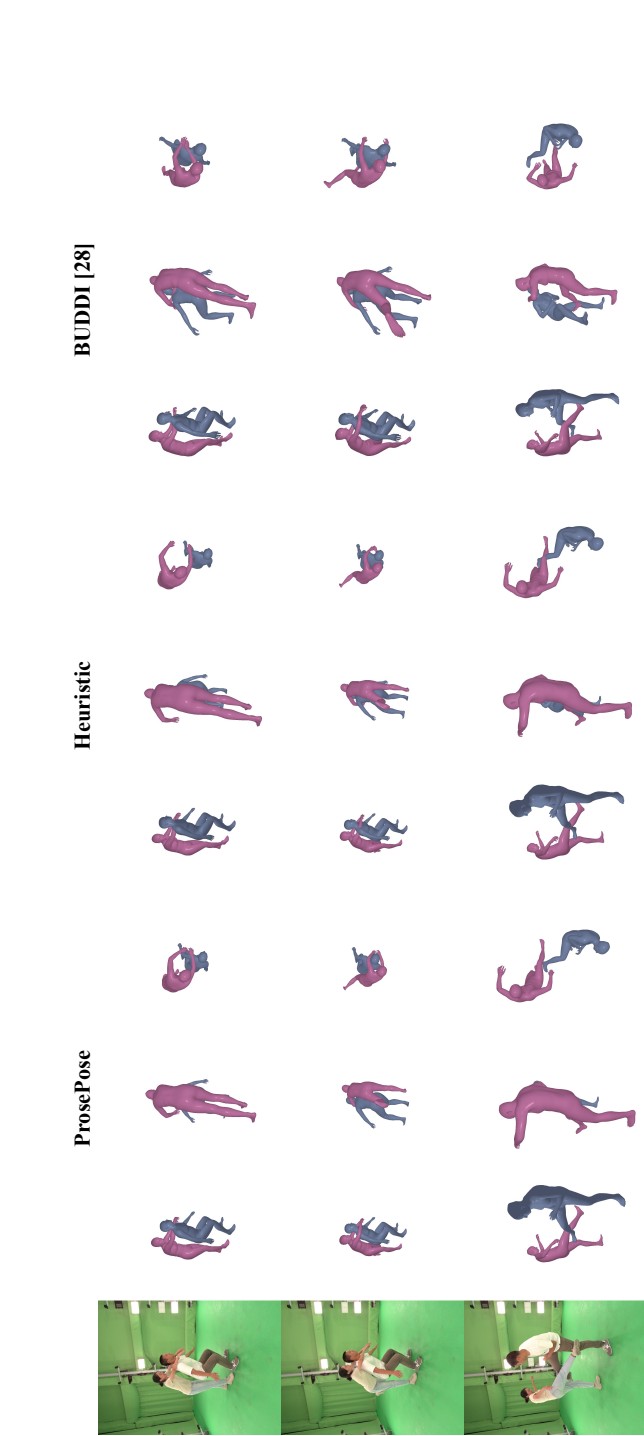

Figure 19: Non-curated examples from the Hi4D test set. They are randomly selected from the examples for which there is at least one non-empty constraint set.

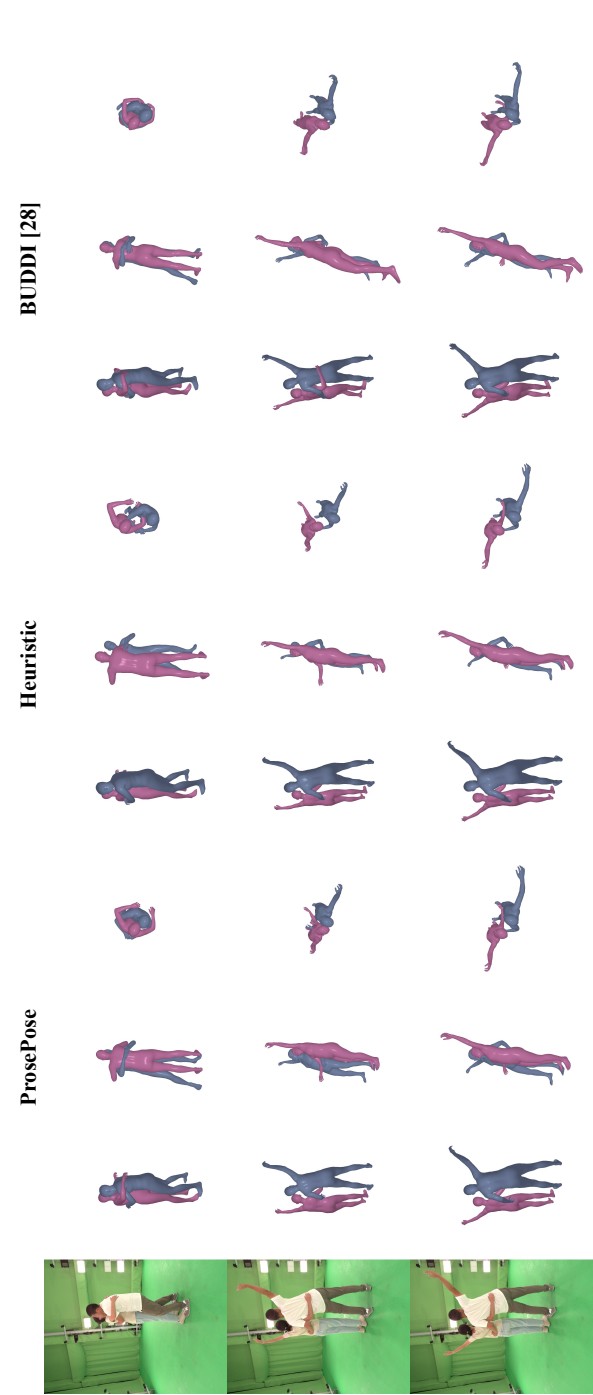

Figure 20: Non-curated examples from the Hi4D test set. They are randomly selected from the examples for which there is at least one non-empty constraint set.

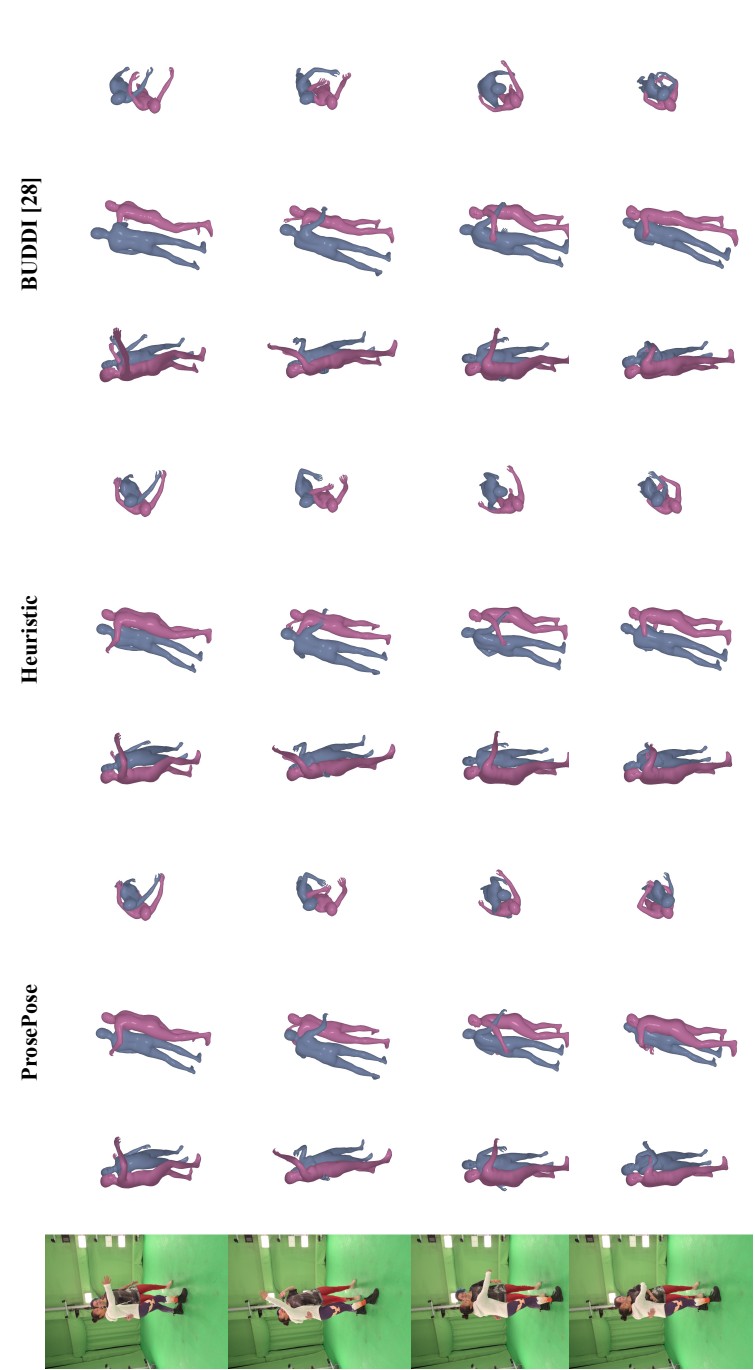

Figure 21: Non-curated examples from the Hi4D test set. They are randomly selected from the examples for which there is at least one non-empty constraint set.

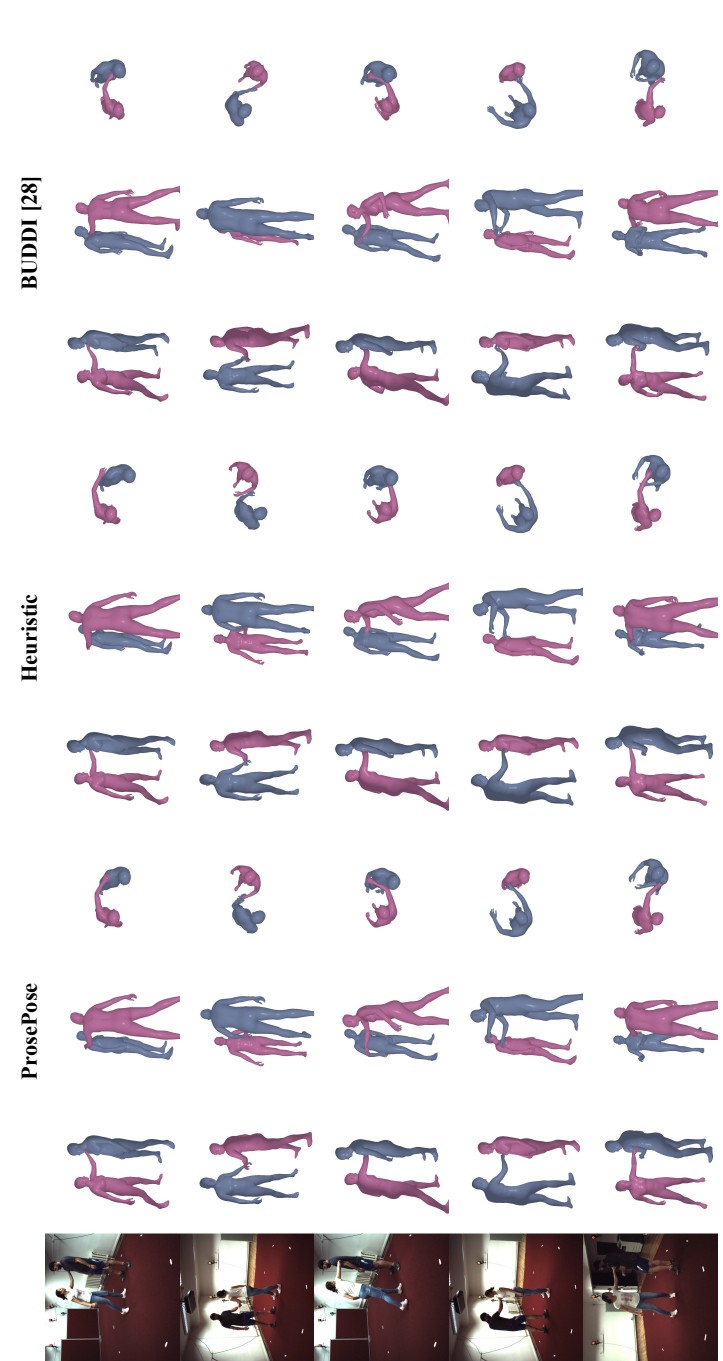

Figure 22: Non-curated examples from the CHI3D validation set (which we use as the test set). They are randomly selected from the examples for which there are at least nineteen non-empty constraint sets (since we set $t = 2$ for CHI3D).

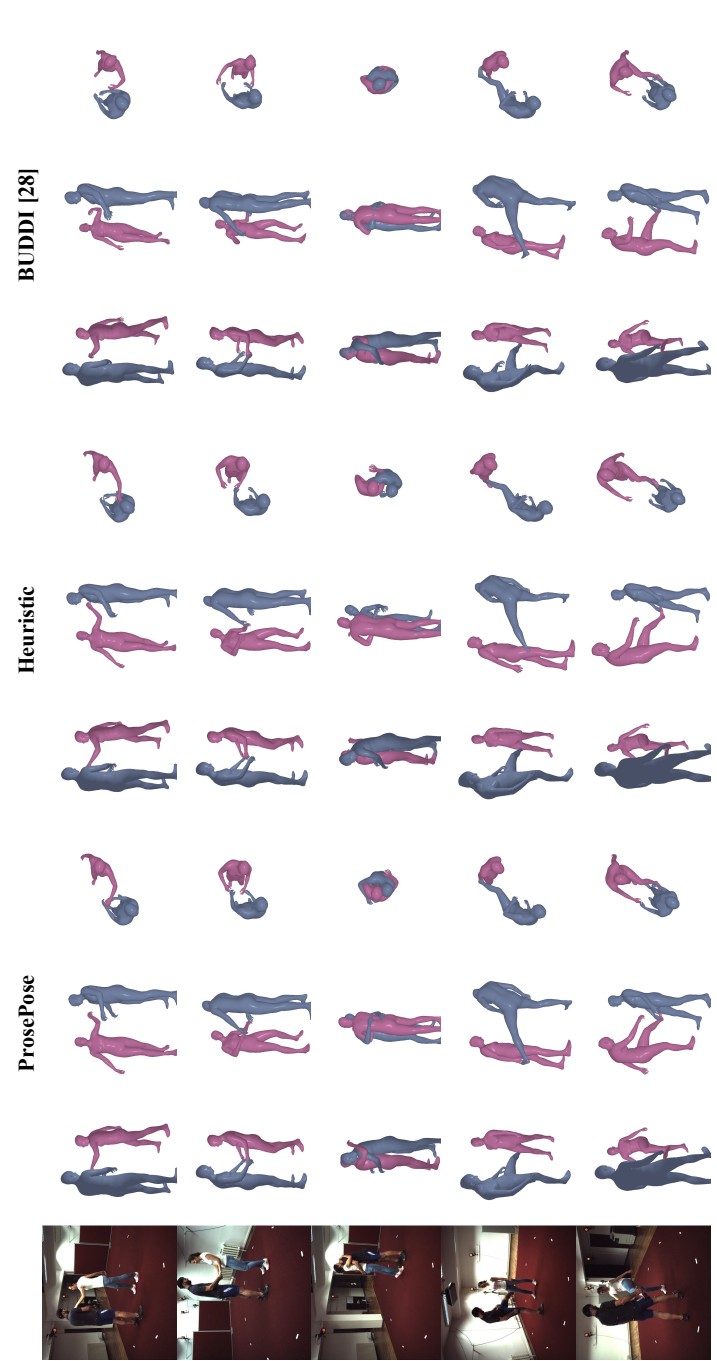

Figure 23: Non-curated examples from the CHI3D validation set (which we use as the test set). They are randomly selected from the examples for which there are at least nineteen non-empty constraint sets (since we set $t = 2$ for CHI3D).

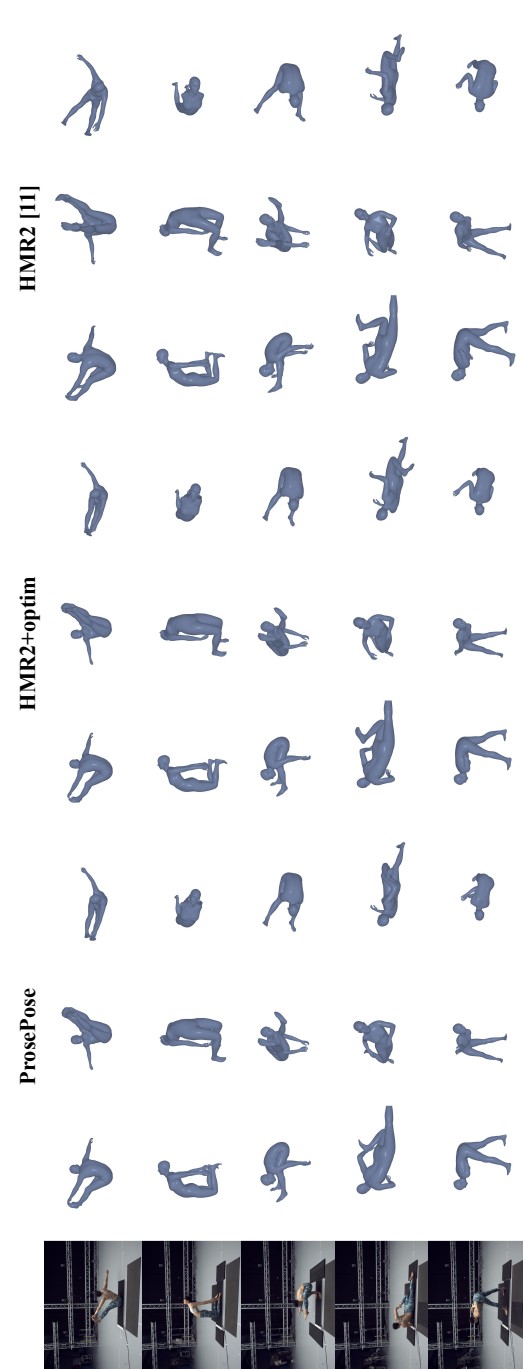

Figure 24: Non-curated examples from the MOYO test set. They are randomly selected from the examples for which there is at least one non-empty constraint set.

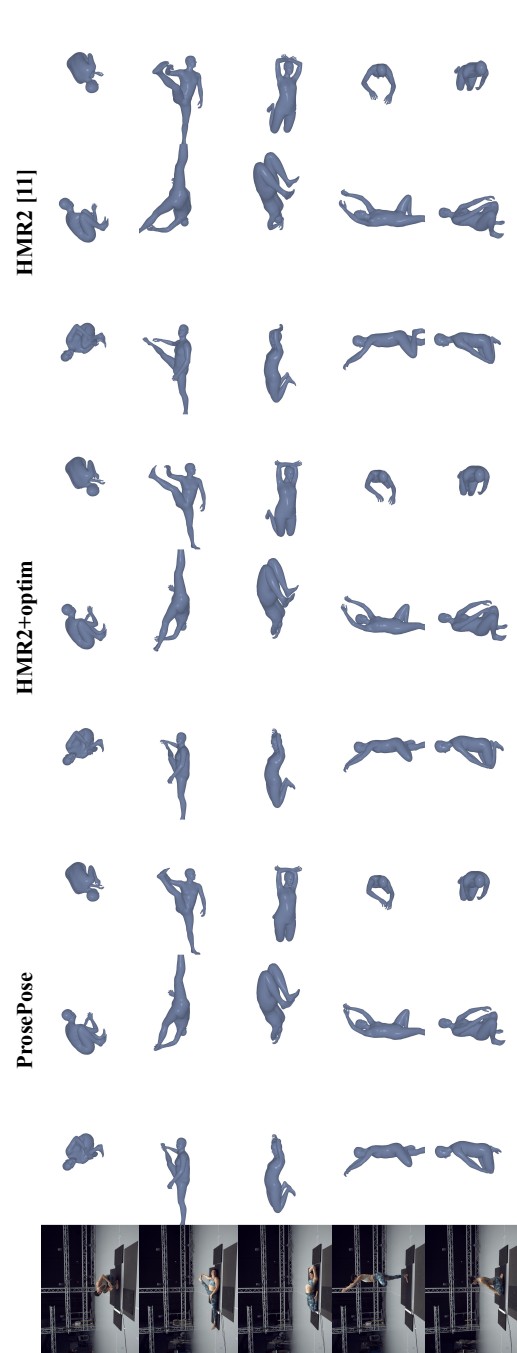

Figure 25: Non-curated examples from the MOYO test set. They are randomly selected from the examples for which there is at least one non-empty constraint set.

