# OpenReview forum: "Pose Priors from Language Models"
_ICLR.cc/2025/Conference — ICLR 2025 Conference Withdrawn Submission_

### Official Review · Reviewer_pTdo · 2024-10-17

**Soundness:** 3
**Presentation:** 2
**Contribution:** 3
**Rating:** 5
**Confidence:** 3

**Summary:**

This paper introduces a 3D pose estimation method leveraging the rich pose semantic prior inherent in LLM. It turns the natural language descriptions of physical interaction into tractable losses for constraining the optimization of 3D pose. The results are interesting, which pose a well estimate of complex scenes with several types of contact between individuals.

**Strengths:**

1. The idea of extracting rich semantic pose prior from LLM to aid pose estimation is interesting and has great potential for future applications, including VR/AR, and metaverse.
2. Experiment results demonstrate the effectiveness of the proposed method.

**Weaknesses:**

1. The organization of the introduction makes it hard to follow.  It looks piecemeal, as the connection and split between the problem to be solved, the motivation, and the main insight are stiff. For example,  in L 56-57, existing methods struggle with solving the proposed problem, but what is the inner reason, why could it inspire the authors to use LLM?
2. The ablation in Table 3 shows that the interpenetration loss (Eq.6) has a negative impact on the estimation results. Is there any possible explanation? Besides, there are no ablation results on PCC, which is also informative in evaluating the effectiveness of both L_LMM and L_p.
3. I wonder if the proposed method can be applied to naturally taken photos without ground truth mesh. A few cases would be intuitive.
4. Since all the losses applied in this method are either unsupervised or constrained on pseudo labels, an error correction mechanism during the optimization may be essential. Once the LLM outputs incorrect contact information, the optimization may lead to distorted mesh.

**Questions:**

Small Suggestions
1. The transitions between sentences could be a little smoother using proper conjunctions for a better reading experience, such as L46-47.
2.. It would be more clear to illustrate by avoiding using super long sentences (L49-52, L76-79).
3. The description of physical contact in the third paragraph of the introduction could be illustrated at L50-51 for better reading.
4. There is a duplicate reference in L687-695.
5. It is unnecessary to put the pose initialization in a sole subsection (Sec 3.2). Considering merging it to the problem definition and placing it in Sec 3.1.

---

### Official Review · Reviewer_zrS5 · 2024-11-01

**Soundness:** 2
**Presentation:** 3
**Contribution:** 2
**Rating:** 5
**Confidence:** 3

**Summary:**

The paper proposes leveraging Large Multimodal Models (LLMs) as a prior to improve self- and inter-person contacts in the task of 3D human pose estimation from RGB images. Specifically, given an image, the LLM is prompted to identify pairs of body regions in contact, which are then converted into vertex-based loss terms. The paper introduces a zero-shot learning setting in which no contact labels are required for training or fine-tuning. This approach extends fitting-based methods by incorporating LLM-guided contact terms into the optimization process. Evaluations on multiple datasets and across two scenarios—single-person self-contact and multi-person inter-person contact—demonstrate performance improvements over the baselines.

**Strengths:**

This is a well-motivated and well-written systems paper. The inclusion of multiple components and stages makes for a challenging setup, yet the paper successfully scales the approach across multiple datasets, achieving performance that is either better than or on par with existing methods. Leveraging LLMs in a zero-shot setting for contact inference is a valuable contribution, automating the process and adding guidance during optimization. This approach appears especially effective in the two-person setting.

**Weaknesses:**

1- Although the proposed zero-shot setting is intriguing at first, its practical applications are challenging to foresee. Despite the claims (line 553), the approach falls short of achieving accurate contact modeling in 3D reconstructions. It’s unclear whether language is a suitable modality to provide sufficient information for this task; in fact, its limitations are inherent. For instance, body segmentations must remain coarse due to natural language constraints, and when coupled with the proposed minimum distance loss between vertex sets (Eq. 3), the null space remains excessively large.

2- While the paper makes a small step in this direction, it lacks insights on how to further improve accuracy. Comparisons with prior work, such as BUDDI, suggest that labeled training data might be more effective for this task, or that the two approaches could potentially complement each other. A stronger setup would demonstrate incremental improvements over each baseline (e.g., BEV+ProsePose and BUDDI+ProsePose, as with Heuristic+ProsePose).

3- Although it addresses a different problem (human-object interaction), a prior work [1] leveraged language to infer contacts between body and object parts, which somewhat diminishes the novelty of this paper.

4- Additionally, a technical limitation lies in the hyper-parameters $N$, $f$ and $t$, which can significantly impact performance. While $N$ is ablated, there is no discussion around $f$ and $t$, or the robustness of the method across images.

[1] Wang, Xi, et al. "Reconstructing action-conditioned human-object interactions using commonsense knowledge priors." 2022 International Conference on 3D Vision (3DV). IEEE, 2022.

**Questions:**

1- Building on the previous comment, I think the arms and legs could be further divided into upper and lower segments, as LLMs should be able to distinguish these finer parts.

2- How consistent are the LLM predictions? In Fig. 5, the plots haven’t yet plateaued, suggesting there may still be room for improvement beyond $N=20$. I’m also curious whether it’s possible to obtain 20 unique responses from an LLM for a single image.

3- What is the proportion of images with an empty constraint set?

4- In Table 3, shouldn’t the setting without the LLM term be equivalent to the “Heuristic” baseline, as mentioned in the text (line 422)?
Also, there seems to be a typo in line 291, where a section number is referenced instead of the correct equation (Equation 3.1).

---

### Official Review · Reviewer_nP8g · 2024-11-03

**Soundness:** 3
**Presentation:** 3
**Contribution:** 2
**Rating:** 5
**Confidence:** 5

**Summary:**

This paper proposes an optimization method to improve the self-contact and close-interaction state of 3D human estimated from images, with the help of LLM contact descriptions. The motivation is that symantic guidance from LLM is very helpful to refine the contact states of single-image 3D human regression. Therefore, starting from 3D human predictions of existing HMR methods ,such as BEV, the proposed method jointly optimize: 1) the contact states based on LLM descriptions; 2) 2D alignment based on 2D keypoints; 3) Gaussian mixture pose prior and L2-norm shape constraints. The experiments are conducted on Hi4D, FlickrCI3D, and CHI3D. Compared with heuristic method, the  proposed method, ProsePose consitently improves PAMPJPE and PCC (percentage of correct contact). While increase the PAMPJPE on CHI3D, compared with baseline method, BEV.

**Strengths:**

1. The idea of using semantic guidance from LLM to refine the contact states of 3D human regression makes sense.
2. The experiment results on Hi4D, FlickrCHI3D, and MOYO look promising.  The qualitative results look very encouraging.
3. The writing is clear and easy to understand.

**Weaknesses:**

1. In Tab. 1, the performance degradation on CHI3D is confusing, because we may draw conflicting conclusion from the conflicting results.
Therefore, to solve this confusing point, we may need further discussion in:
a. a brief explanation of why introducing ProsePose makes the PA-MPJPE on CHI3D get worse.
b. if there is a dataset-specific characteristics that might cause such performance degradation, digging into this point may provide an in-depth understanding of the proposed method.

2. In some qualitative results, refining the contact also rectify the depth ordering between people. Detailed exploration at this might make the proposed method be more attractive. The analysis about this could be performed from two aspects:
a. Quantitative evaluation of how ProsePose might be helpful in improving the depth ordering between people. Especially for the in-the-wild cases, evaluation on Relative human (proposed in BEV) dataset, might be doable during rebuttal, because the original predictions are already provided on its Github page.
b. Qualitative results on challenging InterNet images might attract more readers to use the proposed method.

3. Compared with BUDDI, the advantage is not quite obvious. More explaination about the comparison with BUDDI would be helpful for readers to understand. For instance, discuss and illustrate where ProsePose performs better or worse than BUDDI.

**Questions:**

See the weakness above.

---

### Official Review · Reviewer_vPCs · 2024-11-04

**Soundness:** 3
**Presentation:** 3
**Contribution:** 2
**Rating:** 5
**Confidence:** 4

**Summary:**

This paper presents ProsePose, a zero-shot framework that uses Large Multimodal Models (LMMs) to refine 3D human pose estimates, particularly for poses involving physical contact. The authors leverage the semantic knowledge embedded in LMMs to generate contact constraints from natural language descriptions, which are then used to improve 3D pose accuracy. ProsePose achieves competitive results against state-of-the-art models that require contact annotations, demonstrating its robustness on datasets with both self-contact (e.g., yoga) and person-to-person contact (e.g., dancing, tackling).

**Strengths:**

- The paper is well written and flows well. The terminology and methodology is explained well
- Interesting use of LLMs: With the rise in LLMs, it is interesting to see vision problems being solved via language - especially for 3D computer vision and surpassing traditional methods. The authors have used LLMs to generate constraints for pose estimation.
- The proposed method performs well compared to traditional techniques.

**Weaknesses:**

My main concerns with the paper are:
- It is not clear enough on why solving touch constraints help with better pose optimization. Are there other failure cases in pose optimization than touch constraints. I guess I am not clear on: If I want to use LLMs for better pose estimation - what is the first thing I ll ask LLM. Why is it contact information in this case? Why not some other information say visibility of regions/positioning of regions relative to others.
- What is the practical comparison on solving constraint optimization problem using LLM vs traditional methods. Is it more expensive in terms of compute, costs? Some discussion on this angle will be useful.
- There seem to be a lot of ablations that can be done on the prompting side which seem to be missing in the paper.
  - What are other ways to aggregate N responses from LLM? What if we take the intersection/union of constraints
  - What if I use different temperature for N different calls - will more randomness of LLM in different calls help solve the problem better?
  - How does the method perform if granularity of regions is different - other than SMPL-X
  - What if instead of asking constraints of all regions in one call - you use multiple calls say 10 regions per call. Does it perform any better?

**Questions:**

See weakness

---

### Note · Authors · 2024-11-15

I have read and agree with the venue's withdrawal policy on behalf of myself and my co-authors.